# Potential Role of Curcumin and Its Nanoformulations to Treat Various Types of Cancers

**DOI:** 10.3390/biom11030392

**Published:** 2021-03-07

**Authors:** Md. Tanvir Kabir, Md. Habibur Rahman, Rokeya Akter, Tapan Behl, Deepak Kaushik, Vineet Mittal, Parijat Pandey, Muhammad Furqan Akhtar, Ammara Saleem, Ghadeer M. Albadrani, Mohamed Kamel, Shaden A.M. Khalifa, Hesham R. El-Seedi, Mohamed M. Abdel-Daim

**Affiliations:** 1Department of Pharmacy, Brac University, 66 Mohakhali, Dhaka 1212, Bangladesh; tanvir_kbr@yahoo.com; 2Department of Pharmacy, Southeast University, Banani, Dhaka 1213, Bangladesh; 3Department of Pharmacy, Jagannath University, Sadarghat, Dhaka 1100, Bangladesh; rokeyahabib94@gmail.com; 4Chitkara College of Pharmacy, Chitkara University, Punjab 140401, India; tapanbehl31@gmail.com; 5Department of Pharmaceutical Sciences, Maharshi Dayanand University, Rohtak 124001, India; deepkaushik1977@gmail.com (D.K.); vineetmittalmdu@gmail.com (V.M.); 6Shri Baba Mastnath Institute of Pharmaceutical Sciences and Research, Baba Mastnath University, Rohtak 124001, India; parijatpandey98@gmail.com; 7Riphah Institute of Pharmaceutical Sciences, Lahore Campus, Riphah International University, Lahore 54000, Pakistan; furqan.pharmacist@gmail.com; 8Department of Pharmacology, Faculty of Pharmaceutical Sciences, Government College University Faisalabad, Faisalabad 38000, Pakistan; amarafurqan786@hotmail.com; 9Department of Biology, College of Science, Princess Nourah bint Abdulrahman University, Riyadh 11474, Saudi Arabia; gmalbadrani@pnu.edu.sa; 10Department of Medicine and Infectious Diseases, Faculty of Veterinary Medicine, Cairo University, Giza 12211, Egypt; m_salah@cu.edu.eg; 11Department of Molecular Biosciences, The Wenner-Gren Institute, Stockholm University, S-10691 Stockholm, Sweden; 12Pharmacognosy Group, Department of Medicinal Chemistry, Uppsala University, Biomedical Centre, Box 574, 751 23 Uppsala, Sweden; hesham.el-seedi@farmbio.uu.se; 13International Research Center for Food Nutrition and Safety, Jiangsu University, Zhenjiang 212013, China; 14Department of Chemistry, Faculty of Science, Menoufia University, Shebin El-Kom 32512, Egypt; 15Pharmacology Department, Faculty of Veterinary Medicine, Suez Canal University, Ismailia 41522, Egypt

**Keywords:** *Curcuma longa*, curcumin, anticancer, mechanism of action, cellular mechanisms, nanoformulations

## Abstract

Cancer is a major burden of disease globally. Each year, tens of millions of people are diagnosed with cancer worldwide, and more than half of the patients eventually die from it. Significant advances have been noticed in cancer treatment, but the mortality and incidence rates of cancers are still high. Thus, there is a growing research interest in developing more effective and less toxic cancer treatment approaches. Curcumin (CUR), the major active component of turmeric (*Curcuma longa* L.), has gained great research interest as an antioxidant, anticancer, and anti-inflammatory agent. This natural compound shows its anticancer effect through several pathways including interfering with multiple cellular mechanisms and inhibiting/inducing the generation of multiple cytokines, enzymes, or growth factors including IκB kinase β (IκKβ), tumor necrosis factor-alpha (TNF-α), signal transducer, and activator of transcription 3 (STAT3), cyclooxygenase II (COX-2), protein kinase D1 (PKD1), nuclear factor-kappa B (NF-κB), epidermal growth factor, and mitogen-activated protein kinase (MAPK). Interestingly, the anticancer activity of CUR has been limited primarily due to its poor water solubility, which can lead to low chemical stability, low oral bioavailability, and low cellular uptake. Delivering drugs at a controlled rate, slow delivery, and targeted delivery are other very attractive methods and have been pursued vigorously. Multiple CUR nanoformulations have also been developed so far to ameliorate solubility and bioavailability of CUR and to provide protection to CUR against hydrolysis inactivation. In this review, we have summarized the anticancer activity of CUR against several cancers, for example, gastrointestinal, head and neck, brain, pancreatic, colorectal, breast, and prostate cancers. In addition, we have also focused on the findings obtained from multiple experimental and clinical studies regarding the anticancer effect of CUR in animal models, human subjects, and cancer cell lines.

## 1. Introduction

Globally, cancer is the second leading cause of death and is one of the main causes of public health problems. In 2018 alone, there were about 1.73 million new cancer cases and over 609,000 cancer-related deaths in the United States [1]. Although there are some noticeable advances in cancer treatment, the occurrence of cancer and mortality rate has not decreased in the last 30 years [2]. In the case of treatment and prevention of cancer, improved knowledge regarding molecular changes that contribute to the development and advancement of cancer is crucial. Various common approaches can be used to target specific cancer cells in order to suppress the development of tumor, metastasis, and progression without exerting serious side effects [3]. Along with the chemically synthesized anticancer drugs, various anticancer agents have been extracted from several plants including *Curcuma longa (C. longa)*, *Erythroxylumprevillei*, *Cephalotaxus* species, *Betula alba*, *Catharanthusroseus*, and *Taxusbrevifolia* [2,4]. Furthermore, it has been demonstrated that numerous plant species exhibit anti-cancer properties and there is a growing interest regarding these plants, particularly in developing countries [5,6,7,8,9,10].

Curcumin (CUR) (Figure 1) is a polyphenolic compound extracted from *C. longa* (turmeric) rhizomes. In 1815, this compound was first isolated by two scientists, namely, Pelletier and Vogel [11]. Following this discovery, there was a growing research interest regarding CUR, which led to the identification of the numerous health benefits of CUR. This polyphenolic compound is also familiar as diferuloylmethane. Its molecular weight is 368.38 and its chemical formula is C_21_H_20_O_6_ [12]. CUR has shown its activity against multiple chronic diseases including neurodegenerative disorders, obesity, liver disease, metabolic syndrome, arthritis, inflammation, and multiple cancers [13,14]. Indeed, CUR is a highly effective candidate in cancer treatment as a single-drug therapy or in combination with other therapeutic agents.This natural compound also has the capacity to influence various molecular targets and signaling mechanisms that are linked with various cancers [15,16].

In this article, we have summarized CUR’s anticancer activity against several cancers such as gastrointestinal, brain, head and neck, pancreatic, prostate, breast, and colorectal cancers. Furthermore, we also focused on the findings obtained from several experimental and clinical studies regarding the anticancer action of CUR in animal models, human subjects, and cancer cell lines.

## 2. Mechanism of Action of Curcumin as an Anticancer Agent

An imbalance between cell death and cell proliferation is regarded as one of the major causal factors of cancer [17]. Uncontrolled cell proliferation is likely to occur if the cells skip death, which can lead to various types of cancers [18]. The intrinsic and extrinsic pathways are responsible for generating apoptotic signals. It has been found that the intrinsic pathway plays a role via inducing the mitochondrial membrane to suppress the expressions of B-cell lymphoma-extra large and B-cell lymphoma 2 (Bcl-2) [19]. CUR has the ability to disrupt the balance of mitochondrial membrane potential, which can result in increased Bcl-xL suppression [20]. On the other hand, the extrinsic apoptotic pathway functions via inducing the tumor necrosis factor (TNF)-associated apoptosis and elevating the death receptors (DRs) on cells. In this pathway, CUR plays a role via upregulating DR4 and DR expression [21,22,23]. It has been revealed by in vitro studies that CUR and its derivatives can excellently stimulate apoptosis in various cell lines through downregulating or suppressing intracellular transcription factors. These transcription factors include matrix metalloproteinase-9 (MMP-9), signal transducer and activator of transcription 3 (STAT3), cyclooxygenase II (COX-2), activator protein 1 (AP-1), nuclear factor-kappa B (NF-κB), and nitric oxide synthase [24,25]. CUR can also exert its anticancer effect via reducing lactate production and uptake of glucose in cancer cells through pyruvate kinase M2 (PKM2)downregulation. PKM2 suppression was found to be attained by inhibiting the mammalian target of rapamycin-hypoxia-inducible factor 1α (mTOR-HIF1α) [26]. Multiple analyses have examined the capacity of CUR and derivatives of CUR to inhibit several types of cancers through interacting with various molecular targets (Figure 2).

In CL-5 xenograft tumors, CUR can trigger apoptosis and caused downregulation of cyclin D1, c-Met, Akt, and epidermal growth factor receptor (EGFR) [27]. Furthermore, CUR suppressed metastasis and lung cell invasion via upregulating the expression of HLJ1 in cancer cells [28]. Along with the activity of CUR on nuclear factor-κB (NF-κB) and STAT3 signaling cascades, CUR also suppressed cell cycle arrest and cell proliferation and induced apoptosis by modulating other transcription factors including PPAR-α, Hif-1, Notch-1, β-catenin, p53, Erg-1, and AP-1 [29]. It has been confirmed that CUR suppressed the phosphorylation of focal adhesion kinase (FAK) and increased the expression of multiple extracellular matrix (ECM) components, which further contribute to metastasis and invasion. In a concentration-dependent manner, CUR also increased cell adhesion via inducing various ECM components including fibronectin, laminin, collagen IX, collagen IV, collagen III, and collagen I. Collectively, these findings have indicated that CUR inhibits FAK action via suppression of its phosphorylation sites and triggers ECM components to improve cell adhesion, which can eventually prevent cell migration and detachment of cancer cells. It was reported that suppression of FAK expression resulted in elevated cell adhesion, which eventually playeda role in the anti-invasive activity of CUR [30]. In colorectal cancer cells, CUR decreased the expression of CD24 in a dose-dependent manner. In addition, expression of E-cadherin was elevated by CUR and played a role as a suppressor of epithelial-mesenchymal transition. In colorectal cancer cells, CUR may exhibit its action against metastasis by downregulating CD24, FAK, and Sp-1, and upregulating the expression of E-cadherin [30]. In a study, Zhou et al. [30] assessed the activity of 11 CUR-associated compounds (comprising a benzyl piperidone moiety) in various cancer cell lines. Furthermore, they observed that some of these compounds decreased the level of the phospho-extracellular signal-regulated kinase (Erk)1/2 and phospho-Akt [30]. It has been reported that autophagy and ER stress might have a significant contribution in the case of apoptosis, which is triggered via the CUR analogue B19 in hepatocellular carcinoma cells and the epithelial ovarian tumor cell line, and that suppression of autophagy may elevate CUR analogue-triggered apoptosis via stimulating severe ER stress. In ovarian cancer cell lines, this CUR analogue might also induce apoptosis, autophagy, and ER stress in vitro [31,32]. It was confirmed that autophagy may play role in programmed cell death type II and might effectively inhibit the growth of malignant glioma cells after treatment with CUR [33].

## 3. Bioavailability of Curcumin

CUR is safer to use and it has great potential as an anticancer agent. However, its major drawback is its poor oral bioavailability, which takes place because of its substantial first-pass effect and low aqueous solubility [34,35,36,37,38,39]. At tumor sites, increased permeability and retention action of nanomaterials might ameliorate the buildup of chemotherapeutic agents. In this regard, for instance, micelles, dendrimers, carbon nanotubes, and liposomes, have been utilized as carriers for cisplatin, paclitaxel, doxorubicin, and SN38 in order to decrease adverse effects and improve concentrations of drugs in tumors [39,40,41,42,43,44]. Increased solubility of chemotherapeutic drugs is another benefit of utilizing nanomaterials as drug carriers. There is a growing interest in self-assembling peptide nanofibers due to their easy modification, good biocompatibility, and design flexibility by a “bottom-up” technique [45,46]. These nanofibers have been extensively utilized in several cell cultures and also in drug delivery systems to decrease adverse effects, ameliorate buildup at the tumor site, and improve the solubility of a hydrophobic drug [47]. The enhanced anticancer activity has been observed with the nanofiber-encapsulated antitumor agents including ellipticine, camptothecin, and paclitaxel [48,49,50]. Several studies have confirmed that as drug carriers, the 2-dimensional structure of peptide nanofibers is much better compared to the 3-dimensional structure of nanoparticles. In a study, Wagh et al. [51] showed that peptide-based nanofibershada rapid elimination rate, improved tumor targeting within a shorter period, and better biocompatibility compared to carbon rods, and spherical nanomaterials (selenium and cadmium quantum dots, poly(lactic-co-glycolic acid) or PLGA, cadmium, gold, and polystyrene).

CUR has been widely studied and several synthetic analogues of CUR have been generated and analyzed for potential therapeutic effects [52,53,54,55,56,57,58,59]. Some of these analogues showed excellent actions in several cancer animal models and cell lines. It has been revealed by studies that CUR-associated compounds containing benzyl piperidone possess increased biological effects and absorption [60,61]. In addition, studies have also confirmed effective anticancer activities of CUR analogues [62,63,64,65]. The introduction of CUR into nano-formulations to increase water-solubility has outstandingly transformed its bioavailability. Moreover, nano-formulations have enhanced the transport and improved in vitro CUR levels in the cell, whereas their extended-release formulas as well as their increased compatibility appear to be excellent for their in vivo activities [66,67,68].

## 4. Therapeutic Activity of Curcumin Nanoformulations

Several CUR nanoformulations (Table 1) have been developed so far. Among them, most of these nanoformulations have focused on ameliorating the solubility and bioavailability of CUR and providing protection to the CUR against inactivation via hydrolysis. Some nanoformulations have focused on prolonged retention and circulation in the body, whereas the rest of them concentrated on intracellular release mechanisms and cellular delivery. Multiple nanoformulations of CUR have significant contributions in various pharmaceutical applications and have been demonstrated to have beneficial effects in the diagnosis of multiple human diseases.

### 4.1. Liposomes

Liposomes are spherical vesicles that are composed of multiple or single phospholipid bilayers that closely resemble the structure of the cell membrane [129]. Indeed, liposomes are perfect delivery vehicles for biologically active substances both in vivo and in vitro. There are several benefits of liposomes including greater stability, high biodegradability, and biocompatibility, easy preparation, flexibility, controlled distribution, targeting specific cells, better solubility, and low toxicity [81]. Therefore, liposomes are considered as the most potential drug-carrier system up until now and are preferred by scientists. Liposome’s diameter ranges between 2.5 and 25 mm. Indeed, the size of the vesicle is a crucial factor to estimate the circulation time of liposomes and the amount of drug capsulation in liposomes is affected by the number and size of bilayers [130]. Numerous studies have reported that liposome helps in solubilizing CUR in the phospholipid bilayer and permits CUR to be dispersed over an aqueous medium and elevates the effects of CUR [131]. In addition, the accumulation of liposomal drugs is mainly observed in bone marrow, lung, spleen, liver, or other organs and tissues. This further helps to decrease side effects and ameliorate the drug therapeutic index. Various studies have also found that liposomal CUR is the most appropriate way to provide treatment to multiple cancer diseases. In a study, Dhule et al. [132] reported that liposomal CUR suppressed the growth of the MCF-7 breast cancer cell line and KHOS OS cell line and exerted a potent in vitro and in vivo anticancer effect. In PC-3 human prostate cancer cells, biochemical processes and antitumor efficiency induced via CUR liposomes have been analyzed [133]. The survival rate of CUR liposomes in PC-3 cells was comparatively lower and time-dependent compared to free CUR. Moreover, liposomes promoted the absorption of CUR into the cell, and the period of cell fluorescence intensity was longer and higher in comparison with the control group. Furthermore, expression levels of matrix metalloproteinase-2 (MMP-2)-messenger RNA (mRNA) and its proteins were detected by reverse transcription-polymerase chain reaction and western blotting. It was revealed that expression levels of MMP-2-mRNA and its proteins were gradually decreased along with the rise inconcentrations of CUR liposomes. In PC-3 cells, it was indicated that CUR liposomes mediated intake of drug-loaded liposomes to improve the cytotoxic effects of intracellular drugs. PC-3 cells were simultaneously suppressed via the downregulation of MMP-2 concentrations. In a different study, Tefas et al. [71] developed the liposomes coencapsulating CUR and doxorubicin that decreased the cell proliferation in C26 murine colon cancer and exhibited improved cytotoxic effect compared to its free form. It has been revealed by in vitro study that liposomal CURtreatment resulted in apoptosis [poly (ADP-ribose) polymerase] and dose-dependent growth suppression [3-(4,5-dimethylthiazol-2-yl)-5-(3-carboxymethoxyphenyl)-2-(4-sulfophenyl)-2H-tetrazolium salt] in two human colorectal cancer cell lines (including Colo205 and LoVo cells) [134]. In a similar manner, liposomes coencapsulating resveratrol and CUR exhibited high encapsulation efficiency, polydispersity index, and lower particle size [76]. Interestingly, it was observed that a combination of blue light-emitting diode (BLED) and CUR liposome nanocarriers (LIP-CUR) stimulated photodynamic therapy (BLED-PDT) generated excellent anticancer effect and bioactivity [74]. In addition, they demonstrated that aqueous-soluble F127-CUR (a novel BLED-PDT-based system) has the capacity to mediate CUR’s anticancer effect and facilitate BLED-PDT-mediated apoptosis. Aqueous-soluble F127-CUR markedly elevated the BLED-PDT activity in the cancer cell compared to free CUR. Collectively, these findings suggest that liposomes might be effective as carriers for CUR.

### 4.2. Nanoparticles

The diameter of nanoparticles ranges between 1 and 100 nm, which can be beneficial for drug delivery due to the unique physical, biological, and chemical properties [135]. Indeed, nanoparticles are one thousand times smaller compared to the average human body cell and composed of ingredients that are engineered at the molecular or atomic level. Nanoparticles containing encapsulating drugs have the ability to induce the solubility and pharmacokinetics of drugs, which further offer controlled release and targeted delivery of therapeutic agents [136]. Albumin, gold, magnetic, solid lipid, and polymer-based nanoparticles have been widely used to ameliorate the therapeutic applications of CUR.

#### 4.2.1. Polymeric Nanoparticles

Since polymeric nanoparticles are small in size and biocompatible, these particles can circulate in the blood for a long time [137]. Numerous natural and synthetic polymers have already been identified and used for the production of CUR nanoparticles such as chitosan, hydrophobically modified starch, N-vinyl-2-pyrrolidone, silk fibroin, polyethylene glycol monoacrylate [NIPAAM (VP/PEG A)], PLGA, polyvinyl alcohol, and N-isopropylacrylamide (NIPAAM) [138]. Chang et al. [139] analyzed the molecular processes activated via CUR loaded-poly(lactic-co-glycolic acid) (PLGA) nanoparticles in the case of CAL27 cisplatin-resistant cancer cells (CAR cells). The obtained data indicated that CUR loaded-PLGA nanoparticles regulated the effect of multiple drug resistance protein 1 and the development of reactive oxygen species (ROS) in CAR cells via causing activation of the intrinsic apoptotic mechanism. Compared to native CUR, CUR-loaded PLGA nanoparticlesare more potent in treating CAR cells in conjunction with better in vivo bioavailability and increased in vitro bioactivity. Interestingly, CUR loaded polymeric nanoparticles by means of the Eudragit R E100 cationic copolymer increased the binding and cellular uptake of polymeric nanoparticles, which further increased the cytotoxic effect. Collectively, this formulation of nanoparticles inhibited the growth of tumors and stated 19-times greater growth suppression of colon-26 cells in comparison with the CUR alone [79]. In addition, CUR silk fibroin (CUR–SF) nanoparticles exerted more steady delivery to the colon cancer cells and showed potent anticancer activity compared to its free form in HCT116 cells. It was also summarized that CUR–SFs controlled release has the ability to ameliorate cellular CUR uptake into cancer cells and decrease the cytotoxic effects in normal cells [80]. Khan et al. [140] demonstrated that CUR loaded-PLGA nanoparticles couldsignificantly inhibit increased concentrations of nuclear p65 and hypoxia-inducible factor 1-alpha in lung and breast cancer cells.

#### 4.2.2. Solid Lipid Nanoparticles

Solid lipid nanoparticles contain colloidal submicron particles that are produced via synthetic or natural lipids dispersed in water or aqueous surfactants. Moreover, these nanoparticles are biocompatible, stable, and easily scalable drug delivery systems, along with an increased drug to lipid ratio, which ameliorates the solubility of poorly soluble drugs [141]. It was revealed that solid lipid CUR nanoparticles showed increased solubility compared to native CUR and decreased theeffect of the lipopolysaccharide (LPS)-stimulated pro-inflammatory mediators interleukin (IL)-6, PGE_2_, and NO via hindering activation of nuclear factor kappa B (NF-*κ*B) [142]. In a study, Sun et al. [143] revealed that CUR solid lipid nanoparticles (CUR–SLNs) showed prolonged cellular uptake and obstruction of growth in cancer cells with enhanced chemical stability and dispersibility of the drug. CUR–SLNs were analyzed for their anticancer effects in breast adenocarcinoma cells (MDA-MB-231). Furthermore, CUR–SLNs exhibited increased support and solubility to drug release compared to native CUR. CUR-SLNs also stimulated considerably increased apoptosis in MDA-MB-231 cells. Collectively, these observationssuggest that the use of CUR-SLNs might be beneficial in cancer treatment [141]such as CURC–SLNs combined with doxorubicin and utilized to overcome the Pgp-induced chemoresistance in triple-negative breast cancer cells. Indeed, this formulation was found to be efficient and safe because of its lower toxicity and increased biocompatibility [93]. Wang et al. [144] exhibited through western blot analysis that the CUR–SLNs mediated Bax/Bcl-2 ratio, however, reduced the expression of cyclin-dependent kinase 4 (CDK4) and cyclin D1. These findings indicate that CUR–SLNs might be used as an effective and beneficial chemotherapeutic agent in the treatment of breast cancer [144].

#### 4.2.3. Magnetic Nanoparticles

Magnetic nanoparticles (MNPs) are composed of a metallic oxide core or metal that might be functionalized within an inorganic metal or polymer coating. Furthermore, this coating demonstrates the biocompatibility and stability of the magnetic nanoparticles. They can also be simply manipulated in shape, chemical properties, and size. MNPs contain distinctive physical properties. Moreover, they have a low production cost and are biocompatible with the human body [145]. In the case of pancreatic cancer, Yallapu et al. [146] assessed the in vivo and in vitro therapeutic effectiveness of MNP-CUR formulation. Furthermore, effective internalization of MNP–CUR was observed in human pancreatic cancer cells (Panc-1 and HPAF-II) in a dose-dependent manner, which eventually resulted in effective suppression of growth of Panc-1 and HPAF-II cells in colony formation and cell proliferation assays [146]. In cancer cells, iron oxide nanoparticle core covered by cyclodextrins (CDs) and pluronic polymer (F68) with CUR exhibited increased uptake. It was observed that this formulation suppressed the potential of the mitochondrial membrane and generated more ROS compared to unformulated CUR. Moreover, it exerted a potent anticancer activity along with magnetic targeting capacities and resonance imaging characteristics [85]. In lymphocyte cells, the sustainable delivery of thiolated starch-coated iron oxide nanoparticles comprising CUR showed enhanced compatibility of the system. In cancer cell lines, it showed cytotoxicity because of its increased drug encapsulation, loading efficiency, and stability [87]. In a different study, CUR-loaded Fe_3_O_4_-MNPs exhibited enhanced uptake, which is beneficial for release of the drug in tumor tissues. This formulation was found to be accompanied by imaging applications in tumor tissues [88]. It has been demonstrated that MNPs decorated with PEGylatedCUR(MNP@PEG-CUR) can act as highly biocompatible drug carriers for anti-tumor medicine [89].

#### 4.2.4. Albumin

Indeed, albumin is the perfect material and desirable protein carrier for the delivery of drugs because of its biodegradable, biocompatible, nontoxic, and high binding ability with various drugs. Kim et al. [147] reported that CUR-loaded human serum albumin (HSA) nanoparticles (CUR–HSA–NPs) showed increased in vivo antitumor effect in comparison with the unformulated CUR, with no toxicity in a tumor xenograft animal model. In addition, this analysis indicated that this formulation is an effective drug delivery system for CUR in the cancer treatment. In breast cancer lines, Thadakapally et al. [148] revealed that PEG–albumin–CUR nanoparticles exerted a marked anti-cancer effect, along with better solubility and stable long circulation. In terms of cytotoxicity to Mia Paca-2 cells, Kim et al. [149] reported that paclitaxel/CUR HSA–NPs were effectively internalized into Mia Paca-2 cells and showed a 71% enhancement in IC_50_ versus paclitaxel HSA–NPs. Collectively, these findings indicate that paclitaxel/CUR HSA–NPs can be effectively used as anti-cancer agents in combination therapy [149].

#### 4.2.5. Gold Nanoparticles

Gold nanoparticles possess novel catalytic and optical properties that are biocompatible and non-toxic and have gained substantial interest in various applications. Gold nanoparticles produced with plant extracts are extensively used in various biomedical applications [83]. Indeed, the colloidal stability of these particles retains the physicochemical activities unchanged. Therefore, no alterations will take place in the biological property of the particles.In a study, CUR-encapsulated chitosan-*graft*-poly (*N*-vinyl caprolactam) nanoparticles containing gold nanoparticles (Au–CRC–TRC–NPs) were used for targeted drug delivery and to reveal apoptosis to colon cancer cells [82]. Nambiar et al. [83] produced CUR gold nanoparticles (CUR–AuNPs) by utilizing a cell-culture medium along with or without fetal bovine serum and demonstrated their anticancer activities in human prostate cancer cells. Gold nanoparticles containing CUR (CWAuNPs) were evaluated for their in vitro activities in renal cancer cells. The findings demonstrated that CWAuNPs stimulated apoptosis in the renal carcinoma cell line A498 and were an effective anticancer agent [150]. Similarly, activities of CUR-green generated gold nanoparticles (AuNPs–CUR) were assessed in cell lines of the breast and colon cancers, MCF-7 and HCT-116, respectively. In addition, it was also shown that AuNPs–CUR possesses high apoptotic and antiproliferative activities against cancer cells in comparison with the native CUR [84]. In a different study, Kondath et al. [151] demonstrated the synergistic activity generated via gold core and CUR against breast cancer cells. They also revealed that CUR–AuNPs get coated via proteins in a biological medium, which eventually helps in their endocytosis. Furthermore, within the cells, cAuNPs induced ROS generation, which then subsequently depleted mitochondrial membrane potential. As a result, Bax was released, which activated DNA fragmentation and PARP cleavage [151]. Indeed, these findings suggest the potential of CUR-AuNPs as an effective chemotherapeutic agent.

### 4.3. Conjugates

The complex generated from the combination of two or more molecules, particularly by the covalent bond, is denoted as conjugates. CUR conjugation with hydrophilic polymers and small molecules can raise its solubility as well as oral bioavailability. It has been reported by Manju and Sreenivasan [96] that CUR conjugation with hyaluronic acid reduces the effects of gold nanoparticles (AuNPs) and ameliorates its aqueous stability and solubility [96]. Muangnoi et al. [152] produced CUR-glutaric acid (CURDG) prodrug via ester linkage and evaluated in mouse models. It has been observed that gold nanoparticle–PVP–CUR conjugate (PVP–C–AuNPs) obstructs the aggregation of Aβ(1–6) along with prolonged drug release, increased CUR bioavailability, and loading efficiency (80%) [97]. Piperine (an alkaloid derived from black pepper) is a strong enhancer of CUR bioavailability [153]. Furthermore, this alkaloid plays a role in the brush borders of the intestinal lining, which further resultsin enhanced absorption of the compound. It has also been reported that piperine plays a role in cell metabolism through suppression of cytochrome p450s and UDP glucuronosyl transferases. Moreover, piperine exerts its action on p-glycoprotein [154]. In human or animal models, co-administration of piperine with CUR significantly elevated the serum level of CUR by 2000-fold because of the extensive absorption and bioavailability of CUR with no adverse events [155]. Tang et al. [156] reported that poly-CURs showed cytotoxicity toward cancer cells, however, a polyacetal-based poly-CUR showed increased cytotoxicity toward MCF-7 breast cancer cell lines and the OVCAR-3 and SKOV-3 ovarian cancers. Furthermore, they can be rapidly taken up by the lysosomes of cancer cells, wherein polyacetal-based poly-CUR hydrolyzed and released active CUR. In SKOV-3 cells, it arrested the G0/G1 phase of the cell cycle in vitro and stimulated apoptosis of cells partly via the caspase-3 dependent cascade. In the SKOV-3 intraperitoneal (i.p.) xenograft tumor model, intravenous injection of polyacetal-based poly-CUR resulted in a significant antitumor effect [156].

### 4.4. Cyclodextrins (CD)

Indeed, α-, β-, and γ-CDs are multi-component hybrid, soluble carrier systems that carry non-covalent bound drugs. Cyclodextrins(CDs) are bucket-shaped oligosaccharides composed of 6 (α-), 7 (β-), or 8 (γ-) D-glucopyranose units that are linked via an α-1,4-glycosidic bond to produce macrocycles [103,157]. Interestingly, β-CD, γ-CD, and their derivatives were extensively utilized to deliver the drugs because of their adaptability, relatively easy synthesis, and low price. In recent times, numerous researchers have demonstrated the importance of CD in the CUR delivery system [158]. Yallapu et al. [99] prepared a β-CD facilitated CUR drug delivery system and revealed that β-CD-CUR elevated CUR distribution in prostate cancer cells in comparison with the unformulated CUR and increased its therapeutic value. In a study, Zhang et al. [159] observed that the β-cyclodextrin-CUR (CD15) formulation showed increased cytotoxicity compared to normal CUR via cell cycle arrest and pro-apoptotic activities of lung cancer cells [159]. In addition, experimental observations of this study recommended that CD15 is an effective system for enhancing CUR delivery and its therapeutic efficiency in lung cancer. On the other hand, nanoparticles were developed by utilizing sulfobutyl-ether-β-cyclodextrin, hyaluronic acid, and chitosan and without or with CUR and utilized to treat colorectal cancer and intestinal epithelial cells. It was reported that CUR nanoparticles exhibited increased stability and encapsulation efficiency. Moreover, it reduced the CUR cytotoxicity in normal intestinal epithelial cells and decreased the proliferation of cancer cells [102]. The water-soluble complex of CUR with CD ameliorated solubility and mediated extended release of drugs in retinitis pigmentosa. Interestingly, the findings aided the formulation of eye drops from phytochemicals derived from natural resources [104]. Zhang et al. [159]also assessed the cellular uptake and anticancer effect of CUR-CDs. They observed that CUR-CDs improved delivery of CUR and ameliorated CUR’s in vitro and in vivo therapeutic efficacy compared to free CUR. Thus, via regulating the mitogen-activated protein kinase (MAPK)/NF-κB signaling pathway, CUR-CDs downregulated CyclinE-CDK2 combination, upregulated the p53/p21 signaling cascade, and elevated the expression of Bax/caspase-3 to trigger G1-phase arrest and cellar apoptosis [159]. Collectively, these findings indicate that CUR-CDs might be used to ameliorate delivery of CUR and its therapeutic potential in the case of lung cancer.

### 4.5. Solid Dispersions

A solid dispersion is the molecular dispersion of two different compounds. Typically, it is a hydrophobic drug (specifically, CUR in a solid hydrophilic carrier or matrix) [160]. Solid dispersions are being dissolved as small colloidal particles of any aqueous media in order to release a drug. Moreover, it also reduces the size of the particle to the nano-range along with improved wet ability, which further leads to the increased oral bio-distribution and pharmacokinetic properties of drugs. It is known that solid dispersions are developed via fusion-melt and solvent-based methods, and by joining both solvent and fusion (hybrid) methods [161]. Researchers developed a CUR–Eudragit^®^ PO solid dispersion via a solution mixing method to elevate the stability and solubility of CUR water. In an in vitro transdermal study, the capacity of CUR@EPO as a vehicle to transport CUR in medicinal applications was confirmed. In a different study, a CUR-Gelucire^®^50/13 solid dispersion was developed via spray drying that exhibited improved solubility (3600-fold) in water in comparison with the native CUR. In addition, the anti-inflammatory effect and bioavailability of CUR were greatly ameliorated via solid dispersion because of enhanced gastrointestinal absorption [106]. In a similar manner, CUR solid dispersion-encapsulated temperature-sensitive in situ hydrogels (CSDG) werefound to be effective in treating vaginal bacterial infection via the sustained and stable release of CUR [108]. In a study, Song et al. [162] observed that the solid dispersion efficiently elevated intestinal penetrability and suppressed P-gp activity. In addition, these activities elevated CUR’s anti-proliferative action in MDA-MB-231 breast cancer cells. Following two hours of incubation with CUR, the solid dispersion formulation, and its physical mixture led to the differential cytotoxic activity of paclitaxel via the suppression of the P-gp-induced efflux of paclitaxel in P-gp overexpressing MDA-MB-231 and Lewis lung carcinoma-PK1-P-gp cells. They also summarized that whencompared to CUR, a solid dispersion formulation of CUR with mannitol and D-a-tocopheryl polyethylene glycol succinate might be a promising option to enhancethe oral bioavailability and efficacy of CUR through increased solubility, dissolution rate, cell permeability, and P-gp modulation [162]. 

### 4.6. Micelles

A micelle is termed as a group of amphiphilic surfactant molecules that instinctively aggregate in water into a spherical vesicle. Indeed, it is extensively utilized in delivering poorly water-soluble drugs including CUR [163]. In a breast tumor model, researchers utilized a one-step solid dispersion method to prepare CUR encapsulated polymeric micelles (CUR-M) and evaluated CUR-M efficiency. CUR-M was found to be effective in hindering the spontaneous pulmonary metastasis and growth of breast tumors compared to unformulated CUR [105]. It was observed that the CUR-poly(ethylene glycol) methyl ether (MPEG-PCL) micelle solid dispersion increased the anti-tumor and anti-angiogenesis activities of CUR. The findings of this study also suggested that CUR micelles might be effective in treating pulmonary carcinoma [164]. The outcome of several sizes of CUR encapsulated micelles was evaluated in the in vitro condition for their cytotoxicity in human colon carcinoma cells, intracellular localization, and cellular uptake. The findings also indicated that small-sized CUR-loaded micelles have the efficacy to stimulate increased cytotoxic effects in human colon carcinoma cells in comparison with the larger micelles. Collectively, these findings suggest that uptake/release kinetics, micelle size, and drug loading are vital factors that are essential for nanoparticle drug delivery [113]. As compared to native CUR, it was revealed that CUR loaded into the zein-super hydrophilic zwitterionic polymers, poly(sulfobetaine methacrylate) (PSBMA) micelles showed improved cellular uptake, cytotoxicity, and stability in cancer cells, and pharmacokinetics [115]. Li et al. [165] revealed through the in vitro cytotoxicity assay that CUR micelles were reasonably more efficient compared to native CUR against multiple cancer cell lines because of the improved cellular uptake of CUR, which further resulted in the apoptosis of cancer cell lines. In addition, increased apoptosis of S-65 cancer cells via CUR micelles was observed because of the downregulation of p-Akt, Blc-2, and p-Rb and activation of caspase-9. It was also exhibited that intraperitoneal administration of CUR micelles (25 mg/kg) may markedly suppress the growth of tumors in comparison with the treatment with native curcumin, along with the reduced expression of vascular endothelial growth factor (VEGF) in tumor tissue and markedly elevated apoptosis of tumor cells [165].

### 4.7. Nanospheres and Microcapsules

Nanospheres are referred to as solid matrix particles where the main constituent (drug) is mixed, nonetheless, microcapsules are composed of the outer polymeric shell and internal core. Arunraj et al. [116] developed surfactant-free CUR nanospheres (CNSs) and reported the proof of the anticancer activity of CNSs in breast cancer and osteosarcoma cell lines [116]. In the case of prostate cancer, spherical and smooth CUR encapsulated PLGA nanospheres were found to be effective for clinical applications. It has been concluded by cell viability study that compared to native CUR, CUR encapsulated nanospheres hadthe capacity to further exert a powerful effect against cancer cells [141]. In breast cancer cells, dimethyl CUR encapsulated PLGA nanospheres (ASC-J9) were assessed. PLGA nanospheres were found to be effective in delivering ASC-J9 intracellularly, which is crucial for arresting estrogen-dependent MCF-7 cancer cell growth [166]. Scientists have successfully encapsulated CUR into polyethylene glycol–polylactic acid (PEG–PLA) nanospheres and delivered to MDA-MB-231 and HeLa cancer cells. It was observed that this formulation ameliorated CUR stability and solubility compared to native CUR and exhibited enhanced cytotoxic activities against cancer cells [117]. It was reported that CUR-loaded PLGA nanospheres exhibited strong intracellular uptake of the CNSs in the cells. It was also shownthat CNSs exerted a strong effect on the cancer cells in comparison with free CUR [167]. In the case of all the cancer cell lines, the range for IC_50_ of CUR-loaded PLGA nanoparticles was in between 20 μM and 22.5 μM, whereas this range for free CUR was in between 32 μM and 34 μM. Moreover, this was responsible for a nearly 35% decrease in the IC_50_ value with CUR-loaded nanoparticles. Microcapsules containing mesoporous silica shells and solid lipid nanoparticles were prepared to increase CUR bioavailability [168]. It is an auspicious drug delivery system and more appropriate for poorly soluble drugs. By using the electrospray method, CUR-PLA-based microcapsules were fabricated [169]. Indeed, the study demonstrated outstanding anti-oxygenation and anti-microbial activities and indicated that the PLA-based electrospray technique combined with spherical microcapsules haspotential medicinal applications, mainly in the case of drug delivery.

### 4.8. Nanogels

A nanogel is a nanoparticle (10–100 nm) containing a hydrogel produced via either the chemical or physical cross-linking of polymers under controlled situations. Nanogels’ cross-linked structure provides a powerful base for drug release and storage. It has been observed that it is a possible method to develop and release active forms of drugs to cells to prevent drug immunogenicity, amelioratestability, and maintainthe activity [95]. A colloidal nanogel carrier system was synthesized and studied for the encapsulation of CUR to increase its cytotoxicity and solubility. Indeed, this CUR-nanogel formulation had the ability to destroy the tumor cells compared to CUR alone [170]. A CUR loaded hydrogel nanoparticle was formulated by combining polyvinyl pyrrolidone and hydroxypropyl methylcellulose, which was subsequently evaluated in mouse models for antimalarial activity. It showed the crucial effect of CUR-loaded hydrogel nanoparticles compared to unformulated CUR [171]. Compared to native CUR, CUR loaded into gold nanoparticle-chitosan nanogels exerted enhanced cytotoxic effects in MCF7 and huh7 cell lines [123]. In addition, CUR was delivered as self-assembled capsules with casein nanogels and carboxymethyl cellulose and fabricated with casein and folic acid via the layer-by-layer method. The findings exerted apoptosis, cytotoxicity, and cellular uptake in melanoma cells (MEL-39) [124]. In comparison with the pure CUR and carboxymethyl cellulose and casein nanogels loaded with CUR, 2-folic acid/casein/carboxymethyl cellulose and casein nanogels loaded with CUR had decreased IC_50_ value and exhibited superior cytotoxic effects in MEL-39 cells due to folate-receptor facilitated endocytosis.

### 4.9. Nanodisks

Nanodisks are apolipoprotein-stabilized, self-assembled, and disk-shaped bilayers. In a study, Ghosh et al. [127] first utilized the nanodisk to enhance the CUR’s solubility and targeted release. Nanodisk formulations of CUR were found to be an effective approach to treat mantle cell lymphoma and other cancers [128]. Interaction between glioblastoma multiform cells and CUR nanodisk was mediated via apolipoprotein E primes to elevate CUR uptake and to ameliorate biological activity [172]. As compared to the control CUR, Mangalathillam et al. [173] reported that CUR loaded chitin nanogels exhibited four times more steady-state transdermal flux of CUR. They also revealed through histopathology studies that porcine skin samples that were treated with the prepared materials exhibited loosening of the outermost layer of the epidermis, which further mediated penetration along with no detected signs of inflammation. Collectively, these findings indicate that CUR-loaded chitin nanogels can be particularly used in melanoma treatment through efficient transdermal penetration [173].

### 4.10. Metallo-Complexes

CUR prepared with palladium(II) complexes exerted powerful antitumor action against MCF-7,HeLa, and A549 tumor cells [174]. In this regard, further studies indicated that these complexes triggered tumor cell apoptosis, disturbed mitochondrial membrane potential, and arrested the cell cycle in the S phase via the ROS-dependent cascade. In addition, Vellampatti et al. [175] reported that metallo-CUR-conjugated DNA complexes exerted marked toxic effects toward prostate cancer cells compared to pristine DNA. Furthermore, they evaluated the cellular uptake of these complexes and revealed that DNA complexes containing Cu^2+^/Ni^2+^-CUR showed brighter fluorescence compared to the complexes containing Zn^2+^-CUR.

## 5. Anticancer Activity of Curcumin against Various Types of Cancers

Indeed, CUR has exhibited its excellent activity in inhibiting cancer cell growth and proliferation in different cancers such as gastrointestinal, head and neck, brain, pancreatic, colorectal, breast, and prostate cancers. Several studies have already assessed its activity in animal models and human cell cultures. Furthermore, multiple clinical studies have also analyzed its safety and efficacy (as a monotherapy or in association with other anticancer agents) to treat several different types of cancer. In Table 2, we presentasummary of these clinical studies.

### 5.1. Gastrointestinal Cancers

#### 5.1.1. Oral Cavity and Salivary Gland Cancers

Even though there are very limited studies regarding the activities of CUR in oral cavity cancers, CUR has exhibited promising activities in preventing oral carcinogenesis. CUR alone [193,194] or when taken with piperine [195] markedly decreased oral carcinoma formation in the 7, 12-dimethylbenz[a] anthracene hamster buccal pouch model of carcinogenesis, perhaps because of the CUR’s antioxidant and anti-lipid peroxidative activities along with its activity on controlling carcinogen detoxification. It has been observed that CUR might facilitate anticancer effects via elevating the systemic and local anti-oxidant status, therefore averting DNA damage and lipid peroxidation [196]. It has also been indicated that CUR may play a role as an oral cavity chemopreventive agent because of its capacity to suppress activation of carcinogen via enhancing the expressions and actions of cytochrome P-450 (CYP) 1A1 and/or CYP1B1 (Figure 3) [197]. CUR treatment suppressed the in vitro cell growth via influencing the translation machinery and suppressing the cap-dependent translation in leukoplakia cells (MSK-Leuk1s), instead of mortalized oral mucosa epithelial cells (NOM9-CT) [198]. CUR decreased heat shock protein 70 (HSP70) expression in oral epithelial GNM cells. Increased concentrations of HSP70 protein were found to be linked with the progression of tumors [199]. In a study, Chang et al. [200] revealed that CUR activates p38 in oral keratinocytes, which can eventually activate the transactivator of CCAAT/enhancer-binding protein alpha to trigger insulin-like growth factor binding protein-5 (IGFBP-5). Upregulation of IGFBP-5 is linked tothe inhibition of oral cancer cell tumorigenesis in xenografts in mouse models. CUR showed anti-motility action, which was facilitated through the inhibition of MAPK/ERK and NF-κB signaling and thus downregulation of proteolytic enzymes such as matrix metalloproteinases (MMP)-2/9 and urokinase-type plasminogen activator (uPA) in the invasive oral squamous carcinoma cell line YD-10B [201]. In addition, CUR-retracted smokeless tobacco mediated expression of COX-2 and activation of NF-κB in oral cancer and premalignant cells in vitro [202]. CUR stimulated apoptosis via reactive oxygen species (ROS) generation [203,204,205,206], which indicatesthat CUR can trigger cell death in these cancer cells.

#### 5.1.2. Esophageal Cancer

Squamous cell carcinoma and adenocarcinoma are the two major types of esophageal cancer. Still, the overall survival rate of esophageal cancer patients remains poor with existing therapeutic agents. Therefore, there is a necessity for innovative and effective therapeutic approaches for esophageal cancer, however, only a small number of studies have evaluated whether CUR can be a potential candidate. CUR suppressed NF-κB action and stimulated apoptosis in OE33 and Flo-1 adenocarcinoma cell lines. Moreover, CUR increased cisplatin and 5-fluorouracil-induced chemosensitivity [208]. In a dose-dependent fashion, CUR stimulated cell death in two adenocarcinoma cell lines including OE19 and OE33, along with two squamous cell carcinoma cell lines including KYSE450 and OE21, perhaps via suppressing the ubiquitin-proteasome system [209]. CUR partly reversed the esophageal squamous cell carcinoma growth-associated mitogenic activity of prostaglandin E2 (PGE2) in the squamous cell carcinoma cell line HKESC-1 (Figure 3) [210]. Along with its effective chemotherapeutic activity, CUR may possess chemopreventive activities in esophageal cancer. CUR suppressed the multiplicity and occurrence of preneoplastic lesions when provided during the initiation phase and post-initiation phase in N-nitrosomethylbenzylamine-mediated esophageal carcinogenesis in rat models [211].

#### 5.1.3. Stomach Cancer

Infection of gastric epithelial cells caused by *Helicobacter pylori* is a crucial mechanism in the case of gastric cancer development. Indeed, one of the suggested molecular processes of *H. pylori*-mediated carcinogenesis is the abnormal expression of activation-mediated cytidinedeaminase (AID), which is a mechanism that includes activation of NF-κB via *H. pylori* [212]. CUR was found to downregulate *H. pylori*-mediated AID expression by NF-κB pathway suppression [213]. Furthermore, CUR showed an in vitro and in vivo antimicrobial effect against *H. pylori* and destroyed *H. pylori* [214,215]. Therefore, CUR can be regarded as an effective chemopreventive agent against *H. pylori*-mediated gastric carcinogenesis [213,214,215,216]. A different mechanism that has been associated with chemoresistance and tumorigenesis includes Rho, NF-κB, and Rho effectors rhotekin (RTKN) (Figure 3) [217]. CUR blocked the RTKN-induced anti-apoptotic activity in AGS cells [217], which is a cell line that was formerly utilized to exhibit that CUR has the capacity to suppress the growth of gastric carcinoma cells [218]. CUR suppressed proliferation and invasion via downregulating the action of p21-activated kinase 1 (PAK1) and expression of cyclin D1 in cultured gastric cancer cells (MGC803, MKN1, SGC7901, and BGC823 [219]. Furthermore, in KATO-III gastric cancer cells, decreased levels of cyclin D and E were observed following stimulation of apoptosis via CUR [21]. In the resistant human gastric carcinoma SGC7901/VCR cell line, CUR also has the capacity to reverse the multidrug resistance, which may be linked with a decreased activity and expression of P-glycoprotein (P-gp) and the induction of activation of caspase-3 [220], which further indicatesthat CUR may possess chemotherapeutic activities. In mouse models, chemopreventive activities of CUR have been confirmed in benzo[a]pyrene-mediated for stomach cancer [221] and in N-methyl-N’-nitro-N-nitrosoguanidine-mediated glandular stomach carcinogenesis [222,223,224,225,226,227]. Findings from phase I clinical studies have also indicated biological activities of CUR in cancer chemoprevention [228].

#### 5.1.4. Intestinal Cancer

Anti-cancer activity of CUR has been best described in intestinal cancers by utilizing in vivo animal models and cultured tumor cells. CUR has been found to be better tolerated and its pharmacologically active levels can be attained in colorectal tissues in individuals following its oral administration. However, more robust studies are required regarding the anti-cancer effects of CUR in patients [229,230,231,232]. Chemopreventive activities of CUR have been confirmed in tumor xenografts and in transgenic mice. CUR decreased the suppressed tumor development [224,233,234,235,236,237] or number of abnormal crypt foci [237,238] in the Azoxymethane [224,234,235,236,237,239] and the 1,2-dimethylhydrazine-stimulated [233,238] rat colon cancer model. In addition, CUR treatment reduced tumor growth in HCT-116 colon tumor-bearing mouse models [240]. Various CUR analogs were found to be more potent compared to CUR in some tumor models; for example, GO-Y030 showed greater activity compared to CUR in stimulating apoptosis in cultured human colorectal cancer cells [241] and elevated the lifespan in Apc(580D+) mouse models [241,242]. EF24 markedly inhibited the tumor growth of HCT-116 colon cancer xenografts [57].

Along with chemotherapy, CUR might increase the efficacy of current chemotherapeutics, which might be a better approach in treating gastrointestinal cancers [243]. Combination therapy of dasatinib and CUR was found to be extremely effective in triggering regression of intestinal adenomas in APC (Min/+) mouse models [244]. In comparison with resveratrol alone, a combination of resveratrol and CUR effectively suppressed the growth of HCT-116 cells both in vivo and in vitro in SCID xenografts [245]. CUR sensitized colorectal cancer to the anti-metastatic and anti-tumor activities of capecitabine via inhibiting the NF-κB signaling pathway in orthotopic colorectal cancer cell xenografts [246]. Along with oxaliplatin, liposomal CUR markedly suppressed the growth of Colo205 and LoVo xenografts and showed anti-angiogenic activities [134]. CUR also enhanced the pro-apoptotic activities of oxaliplatin [247,248], 5-fluorouracil [247,249], and sulindacsulfone [250] in cultured colon cancer cells. Numerous molecular targets of CUR have already been detected in colon cancer cell lines, along with its well-defined activity in inhibiting the COX-2 and NF-κB signaling pathway as well as other recognized processes of stimulating apoptosis (Figure 3) [251,252,253,254,255]. These involve suppression of signaling pathways including Ca^2+^/calmodulin [256], c-jun N-terminal kinase (JNK) [257], Akt/mTOR [258], IGF-1R [247,259], Wnt/β-catenin [260,261,262], and EGFR [232,244,248,249]. CUR might inhibit the expression of factors containing procarcinogenic activities including neurotensin-mediated expression of IL-8 [263]. Moreover, both in vitro and in vivo studies have revealed that CUR inhibited the proteasome effect and stimulate apoptosis in human colon cancer cells [240].

Radiotherapy is a crucial part of the preoperative treatment for rectal cancers. CUR has the capacity to be an effective radiosensitizing agent with increased anti-angiogenic activities in radiotherapy for colon cancer [253]. In multiple colon cancer cell lines, CUR blocked the transient inducible NF-κB signaling pathway that provided a pro-survival response to radiation [254]. Moreover, CUR sensitized HCT-116 xenografts to gamma-radiation [252], which indicatesthat CUR administration might be utilized to decrease the side-effects that are commonly observed with radiation therapy. Chemopreventive activity of CUR in familial adenomatous polyposis (FAP) has been found to extend the opportunity for cancer prevention and intervention. The combination of quercetin and CUR reduced the number and size of rectal and ileal adenomas in FAP patients [264]. Dietary CUR decreased COX-2 expression and exhibited antioxidant properties that have been linked with hindrance in adenoma development in the Apc (Min/+) mouse model of FAP [265,266]. Although these initial findings are promising, further investigation is required regarding the chemopreventive activities of CUR in FAP.

#### 5.1.5. Hepatic Cancer

Only fewer individuals with hepatocellular carcinoma are diagnosed at early stages and therapeutic options are highly inadequate. At present, surgery is considered as the most effective therapeutic strategy. Nonetheless, there is poor long-term survival, and recurrence rates are high. Usage of chemotherapy and radiation is limited and there is a strong need todiscover novel chemotherapeutic and chemopreventive agents. Multiple experiments have evaluated the in vitro anti-carcinogenic action of CUR in hepatic cancer cells. GL63 (a CUR analogue) suppressed the growth more effectively compared to CUR in HepG2 cells. Furthermore, this finding was linked with the activation of apoptosis and ER stress, an effect that was not seen with CUR [267]. Interestingly, CUR stimulated apoptosis and G2/M arrest in multiple hepatoma cell lines including QGY-7703, SK-Hep-1, Hep3B, and Huh7, however, HepG2 cells were found to be CUR-insensitive [268]. Nonetheless, various other analyses stated the activities of CUR in HepG2 cells such as stimulation of apoptosis [269,270,271] via mitochondrial DNA damage and mitochondrial hyperpolarization [269,270] or via a p53-dependent pathway [271], and suppression of hypoxia stimulated angiogenesis by downregulation of hypoxia-inducible factor HIF-1 [272]. Knockdown of the extracellular matrix metalloproteinase inducer EMMPRIN (CD147) showed sensitivity toward CUR [273] and CUR blocked the c-Met promoter transactivation via activated protein 1 (AP-1) in Hepa1-6 cells [274]. CUR showed a marked anti-invasion effect thatwas linked with the suppression of metalloproteinase MMP-9 in the extremely invasive SK-Hep-1 hepatocellular carcinoma cell [275]. CUR alone or in combination with doxorubicin or cisplatin showed apoptotic and cell growth inhibitory activities, partly because of the alterations in NF-κB concentrations in hepatic cancer HA22T/VGH cells [276].

CUR exerted anticancer activities in multiple in vivo hepatocellular carcinoma models. CUR markedly suppressed the formation and growth of liver adenoma in mouse models receiving N-bis (2-hydroxypropyl) nitrosamine [277]. In addition, CUR decreased tumor multiplicity and occurrence in mice with diethyl nitrosamine-mediated hepatocarcinogenesis [278]. Even at the highest doses, tetrahydro curcumin did not exert any cytotoxic effect in HepG2 cells, however, it exerted anti-angiogenic activities in HepG2 xenografts [279,280,281]. In an orthotopic implantation model of hepatocellular carcinoma CBO140C12 cells, CUR inhibited intrahepatic metastasis, but did not influence the growth of the implanted tumor [282]. In Wistar rats, CUR exerted beneficial activities against oxidative stress during chemically triggered hepatocarcinogenesis in N-nitroso-diethylamine-introduced and phenobarbital-mediated hepatocarcinogenesis [283,284,285], which indicate that CUR might play a role as an effective therapeutic agent in liver cancer.

#### 5.1.6. Pancreatic Cancer

Pancreatic cancer is frequently detected in an advanced stage and is characterized by increased invasiveness, rapid disease progression, and resistance to chemotherapy. In pancreatic cancer, CUR may have antitumor activity alone [286,287] or in association with other therapeutic agents including celecoxib [288] or gemcitabine [289,290,291,292]. When a polymeric nanoparticle-encapsulated CUR was administered systematically, it blocked metastasis and tumor growth in subcutaneous and orthotopic Pa03C xenograft models of pancreatic cancer [286], while liposomal CUR reduced tumor growth in MiaPaCa-2 subcutaneous xenografts [293] and in cultured pancreatic cancer cells [294]. Interestingly, nanoparticle encapsulation of CUR into a MePEG/poly-epsilon caprolactone (PCL) diblockcopolymeric micelle increased the cytotoxicity and uptake in the pancreatic cancer cell lines PANC-1 and MiaPaCa-2 [295] and the polyethylene glycosylated (PEG) CURconjugate was more effective in inducing apoptosis and cell cycle arrest compared to free CUR [296]. In phase II studies, CUR was found to be better-tolerated and exerted a biological effect in some advanced pancreatic cancer patients [34].

Suppression of the NF-κB signaling pathway via CUR has been reported in the case of pancreatic cancer [297,298,299]. NF-κB suppression was reliant on decreasing various transcription factors including Sp1, Sp3, and Sp4, and was associated with ROS stimulation [297,300]. Various other targets of CUR involve the transcription factor Wilms’ tumor gene 1 (WT1) [301], heme oxygenase-1 (HO-1), ATM/Chk1 [302], survivin/BIRC5 [303], STAT3, IL-8 receptors CXCR1 and CXCR2 [304], Notch-1 [305], EGFR [306], miR-21and miR-22, micro RNAs miR-200, PGE2, COX-2, and Akt (Figure 3). The activity of STAT3 was also suppressed via GO-Y030 (a potent analog of CUR) [307] in pancreatic cancer cell lines. Indeed, other CUR analogs exhibited enhanced potency compared to CUR in the case of pancreatic cancer. In comparison with CUR alone, FLLL11 and FLLL12 markedly induced apoptosis and suppressed cell viability [308]. CDF (a fluorocurcumin analog) exhibited substantially higher bioavailability in pancreatic tissue compared to CURalone [309].

### 5.2. Head and Neck Cancer

Globally, head and neck squamous cell carcinoma (HNSCC) is considered as the sixth most common form of cancer, and every year, over 30,000 cases of HNSCC are detected. In general, HNSCC arises in the pharynx, larynx, paranasal cavities, and oral cavity. It has been confirmed by in vitro studies in various cell lines of head and neck cancer that CUR has the capacity to suppress cell growth because of its activities on several cellular mechanisms associated with cell proliferation (especially STAT3 and NF-κB), which have been detected to be overexpressed in various head and neck cancers. Furthermore, CUR can also cause NF-κB downregulation and suppress the interleukin-6 (IL-6)-induced STAT3 phosphorylation, which can eventually result in the suppression of cancer cell proliferation [186].

In a study, Kim et al. [186] estimated the activity of CUR on suppressing the proinflammatory cytokines and IκB kinase β (IκKβ) activity in HNSCC patients. These patients were treated with chewable CUR tablets (2 mg), after that, saliva samples of the patients were collected before and after administering the chewable CUR tablets. In these saliva samples, IκKβ activity and levels of salivary cytokines such as IL-6 and IL-8 were measured. Furthermore, CUR reduced the activity of IκKβ in the salivary cells of HNSCC individuals. There was a small decrease in the expression of IL-8 in eight out of 21 post-CUR samples. A significant decrease was also observed in the expression of various other cytokines, for instance, IL-2, IL-12p70, IFN-γ, and IL-10 clustered together, and also *tumor necrosis factor-alpha (TNF-α)* and granulocyte-macrophage colony-stimulating factor clustered together. Since these findings suggest the suppressive activity of CUR on IκKβ action in the salivary cells of individuals with HNSCC, it has been suggested that IκKβ can be considered as a biomarker for identifying the activity of CUR in head and neck cancer [186].

### 5.3. Glioblastoma and Brain Cancer

In humans, glioblastoma (GBM) is considered the most common type of malignant brain cancer and responsible for around 15% of all CNS tumors [310,311]. The use of radiation therapy and surgical intervention to treat GBM and brain tumors is restricted because of the infiltration of cancer cells into the healthy brain, which can eventually exert harmful effects following treatment [312]. Thus, there is a growing interest in alternative therapies by means of naturally occurring compounds including CUR due to their fewer side effects compared to conventional therapies. CUR can exert effects in various molecular targets, thus fighting against the brain tumors that might need multiple cellular mechanisms, for instance, metastasis, invasion, angiogenesis, autophagy, and apoptosis. Blood–brain barrier (BBB) penetration is regarded as the rate-limiting step for numerous anti-cancer drugs, however, CUR showed the capacity to penetrate BBB at increased concentrations [313]. It has been revealed by an in vivo study (involving human glioma U-87 cells xenografted into athymic mouse models) that CUR has the capacity to inhibitglioma angiogenesis via downregulating endothelial cell markers (i.e., CD105 and CD31 mRNA) and suppressing MMP-9 [313]. In U-251 malignant glioblastoma cells, CUR mediated G2/M cell cycle arrest was explained by the elevation of the protein kinase 1 (DAPK1) level, which suggests that inhibiting DAPK1 via CUR stimulates cell arrest and also causes NF-κB and STAT3 suppression and caspase-3 activation [314].

### 5.4. Breast Cancer

Breast cancer is a major cause of death in women [176]. It has been revealed by the meta-analysis of 21 retrospective studies that despite endocrine therapy, chemotherapy, radiation therapy, and lumpectomy, the occurrence rate of breast cancer is still high [315]. Thus, there is a need for more effective therapeutic approaches. In MCF-7 breast cancer cells and MCF-10A human mammary epithelial cells [176], an excellent reduction in telomerase activity was noticed due to CUR treatment in a concentration-dependent manner, which was found to be associated with hTERT downregulation via CUR instead of the c-Myc mRNA pathway [176]. In BT-483 and MDA-MB-231 breast cancer cell lines, the activity of CUR on NF-κB, matrix metalloproteinases, and cell-cycle regulatory proteins was also assessed. CUR has been reported to downregulate NF-κB, which can further result in antiproliferative activity. Nonetheless, a reduction in CDK4 BT-483 and cyclic D1 in MDA-MB-231 cells was also noticed with CUR treatment. Treatment with a combination of CUR and arabinogalactan induced apoptosis via disturbing the mitochondrial membrane, elevating the levels of ROS, and reducing glutathione in the MDA-MB-231 cell line. Furthermore, CUR also suppressed breast tumors through *p53*gene overexpression and via decreasing the levels of antigen ki-67. In a different study, CURalso suppressed the levels of inflammatory cytokines CXCL1/2 in MDA-MB-231 cells. Moreover, suppression of CXCL1/2 via CUR also suppressed the expression of multiple metastasis-promoting genes including chemotactic receptor CXCR4. It was also reported that by suppressing various types of steroid receptors, dimethyl CUR (ASC-J9) is effective atfighting against estrogen-dependent breast cancer [176].

In a clinical study, in 14 individuals with metastatic or locoregionally recurrent advanced breast cancer, Bayet-Robert et al. [176] assessed the tolerability and feasibility of the combination of docetaxel and CUR. Intravenous infusion of docetaxel (100 mg/m^2^) was administered to participants every three weeks for six chemotherapy cycles and oral administration of CUR (starting dose was 500 mg/day and the dose was elevated until a dose-limiting toxicity was observed) was continued for seven consecutive days in each cycle (from five days before to two days following docetaxel administration). In the participants, the primary outcome was the maximum tolerated dose of CUR when taken in combination with a standard docetaxel dose. On the other hand, secondary endpoints were a clinical response, safety, and toxicity to the combination therapy, along with levels of a marker for mCEA tumor and VEGF as a positive endogenous modulator of angiogenesis [176].

Indeed, 8 g/day as in higher doses was the maximum tolerated dose of CUR. Furthermore, various dose-limiting toxicities including severe diarrhea, anemia, and neutropenia were seen, which resulted in the termination of the clinical trial in two participants. Other toxic effects including fatigue, conjunctivitis, dermal changes, nail changes, hand-foot syndrome, and oral cavity mucositis were either not continuous or were easilytreated, and so therefore did not influence the trialcontinuation. Due to the noncompliance of multiple individuals with doses over 6 g/day, this dose was suggested as the maximum tolerated dose to be regardedfor phase II clinical trials. To some extent, the improvement was observed in most of the participants in cases of clinical and biological responses. Furthermore, in that study, three participants showed at least six weeks following the last cycle of treatment. However, no advancementin the disease was seen in any of the participants. It was observed that the combination of CUR/docetaxel markedly reduced the VEGF levels following three cycles of treatment [176].

### 5.5. Colorectal Cancer

Colorectal cancer is a very common form of malignant cancer [244]. In addition to chemotherapy, tumor tissue was surgically removed from the individuals with colorectal carcinoma, however, over half of the individuals suffered from relapses [316]. In malignant colorectal cells, CUR treatment reduced the levels of M_1_G without altering the levels of COX-2 protein [180]. Furthermore, administration of CUR downregulated the miR-21 gene (which is found to be overexpressed in colorectal cancer cells) via suppressing the binding of activator protein 1 (AP-1) with miR-21 promoter [317]. In HCT116 colorectal cancer cells, CUR treatment led to cell cycle arrest in the G_2_/M phase via miR-21 gene regulation and suppressed the growth of tumor tissues [317]. It has been confirmed that enhanced response to radiation therapy can be obtained by combining CUR in the treatment because CUR has the capacity to target NF-κB [252]. In a different study, the inhibitory activity of CUR was increased against colon cancer cells by combining CURwith ERRP (an inhibitor of pan-ERBB) [318].

Sharma et al. [179] in a dose-escalation study evaluated the pharmacological activity of CUR in 15 individuals with advanced adenocarcinoma of the colon or rectum refractory to standard chemotherapies. For up to four months, participants of the study received various doses of oral CUR ranging from 0.45 to 3.6 g/day. After that, levels of CUR and its metabolites were also assessed in plasma, feces, and urine. Furthermore, oxidative DNA adduct (M_1_G) levels, the activity of glutathione S-transferase (GST), and the extent of ex vivo stimulation of PGE2 in patient blood leukocytes were evaluated as biomarkers of CUR function. It has also been found that intact CUR and its glucuronide and sulfate conjugates were identified in plasma at a concentration of 10 nmol/L and also in urine [179]. Indeed, no dose-limiting toxic effects were identified. Interestingly, neither any activity on basal PGE2 levels in leukocytes was seen following CUR treatment at any of the doses, nor was any alterations observed in the LPS-mediated PGE2 generation at doses in between 0.45 and 1.8 g/day. CUR treatment at the dose of 3.6 g/day resulted in a 62% and 57% decrease in the inducible levels of PGE2 in patient blood samples one hour following treatment on days 1 and 29, successively, in comparison with the baseline levels. M_1_G levels and total GST activity in leukocytes exhibited significant differences between participants, however, no treatment-associated activities were noticed. According to these findings, researchers indicated an oral dose of CUR 3.6 g/day for a phase II trial in cancers in sites outside the gastrointestinal tract thatneed systemic actions [179].

In another study, Garcea et al. [180] measured the pharmacological activity of CUR in the colorectum by COX-2 and M_1_G levels in 12 individuals with colorectal carcinoma after oral treatment of CUR at doses of 450 mg, 1800 mg, or 3600 mg per day. Biopsy and blood samples of the malignant and normal colorectal tissue were obtained from the individuals at designated time intervals and examined for the concentrations of CUR, COX-2, M_1_G, and CUR metabolites (CUR glucuronide and CUR sulfate). Increased CUR levels were found in normal compared to malignant colorectal tissues of individuals receiving CUR at a dose of 3.6 g per day, along with a trace amount of CURin the peripheral blood circulation. In addition to this, metabolites of CUR were also identified in the colorectum of these individuals [180]. Baseline levels of M_1_G were 2.5-times higher in malignant tissue in comparison with the normal tissue in the same group of individuals, which were markedly decreased following CUR administration. However, COX-2 levels in malignant colorectal tissue were not decreased via CUR. Collectively, these results indicate that CUR administration at a dose of 3.6 g/day can achieve pharmacologically active levels in the colorectum with minimum distribution outside the digestive tract [180].

Various researchers have studied the mode of action of the anticancer effect of CUR in colorectal cancer. In a dose-escalation pilot study, Plummer et al. [178] evaluated the activities of *C. longa* extract (comprising desmethoxy curcumin and CUR) on the suppression of the COX-2 effect and therefore the PGE2 levels in 15 individuals with advanced colorectal cancer. Participants of the study were divided into five groups receiving once-daily oral curcuminoid doses between 40 and 200 mg for a minimum of 29 days. A comparisonofthe PGE2 concentrations in blood samples obtained from participants exhibited a noticeable difference between participants in different groups and reduced PGE2 concentrations with an elevated CUR dose, which evidently suggests dose-dependent CUR-mediated COX-2 inhibition [178]. In a clinical trial, this was further analyzed by Carroll et al. [182] to evaluate the activities of CUR when administered orally in order to prevent colorectal cancer. In that study, 44 smokers with eight or more aberrant crypt foci (ACF) on diagnosing colonoscopy were involved and were divided into two groups receiving oral CUR at a dose of either 2 g or 4 g/day for 30 days. 5-hydroxyeicosatetraenoic acid (5-HETE) and PGE2 levels within ACF were evaluated, along with the decrease in the number and/or proliferation of ACF (estimated via rectal endoscopy and Ki-67 immunohistochemistry assay, successively). The decrease inACF was utilized as an estimation of the cancer-preventive efficacy of CUR, supposing that decreasing the levels of 5-HETE and PGE2 in the colorectal mucosa would lead to decreased ACF formation and epithelial crypt proliferation [182].

Any decreased levels of 5-HETE or PGE2 within normal mucosa or ACF were seen with any doses of CUR and there was alsoa decreased Ki-67 level in normal mucosa. Furthermore, there were no alterations in the number of ACF in the group treated with 2 g of CUR. Nevertheless, a marked decrease in the number of ACF was noticed in the group receiving 4 g of CUR, which was found to be linked with a marked rise in the plasma levels of CUR conjugates, which further suggeststhe action of systematically delivered CURconjugates on the decrease inACF number instead of the locally delivered CUR [182]. In a study, in individuals with colorectal cancer, He et al. [181] studied the activities of CUR on *p53* expression in the colorectum tissue and the serum TNF-α levels. In total, 126 colorectal cancer patients were randomly divided into two groups receiving either the placebo or CUR (at an oral dose of 360 mg, 3 times/day) during the period ahead of surgery. Blood and colorectal biopsy samples were obtained from the participants before and after treatment and were studied for serum TNF-α levels and *p53* expression, respectively. A marked decrease in the serum TNF-α level was detected in the participants receiving CUR, while no such activity was found in the placebo group.A number of apoptotic cells were also elevated following CUR treatment in comparison with the baseline values, while no noticeable alteration was detected in the placebo group. In addition, CUR treatment elevated the expression of Bax and *p53* and suppressed theBcl-2 expression in the colorectal tissue [181].

In a clinical trial, Cruz-Correa et al. [185] evaluated the safety and efficacy of CUR in familial adenomatous polyposis. In that clinical study, 44 participants with familial adenomatous polyposis with a minimum of five intestinal adenomatous polyps who had not gone through colectomy were incorporated into that study and were randomly divided into two groups receiving either the placebo or pure CUR (oral dose of 3 g/day) for 12 months. In addition, main endpoint measures including the size and number of lower gastrointestinal tract polyps were evaluated every four months for one year. No marked difference was observed in the mean size or mean number of polyps between the placebo group and CUR group at the end of the study. Furthermore, adverse events were very uncommon and not considerably different from the placebo group. In individuals with familial adenomatous polyposis, these findings suggest the low efficacy but high safety of oral CUR at the administered dose [185].

### 5.6. Prostate Cancer

According to a recent report of the American Cancer Society, around 2.9 million males have been diagnosed with prostate cancer in the United States [319], which makes this cancer the second main cause of cancer-related death in males [320]. CUR has exhibited a powerful capacity to stimulate apoptosis and suppress proliferation in prostate cancer both in vitro and in vivo [321] by affecting various cellular mediators including NF-κB, EGFR, and MAPK [322,323]. In a study, CUR was found to have the capacity to cause protein kinase D1 (PKD1) activation, which can lead to the weakening of the oncogenic signaling via MAPK and β-catenin [324] and subsequent suppression of prostate cancer development [324]. Furthermore, there was a significant PKD1 downregulation after progression from androgen-dependent to androgen-independent prostate cancer [324], which influencedthe motility and invasion of prostate cancer through interaction with E-cadherin [317]. Thus, it has been regarded as a new therapeutic target for cancer in general and particularly for prostate cancer [325]. Along with CUR, some of the CUR derivatives have also exhibited anti-cancer effects against prostate cancer. Interestingly, metallo-CUR conjugated DNA complexes exerted marked toxicity to prostate cancer cells (DU145, LNCaP, TRAMP-C1, 22Rv1, and PC3) [175]. In androgen-dependent prostate cancer, dimethyl CUR(ASC-J9) exhibited good activity in enhancing androgen receptor degradation [326,327].

In a clinical study, Hejazi et al. [189] evaluated the activity of CUR on the oxidative status of individuals with prostate cancer during radiotherapy. In that clinical study, 40 participants were incorporated in the trial and were randomly assigned to receive either placebo or oral curcuminoids (CUR, bisdesmethoxycurcumin, and desmethoxycurcumin, 3 g/day) before and during external-beam radiation therapy. Three months after radiotherapy, the outcome measures of oxidative status were the plasma total antioxidant capacity (TAC), glutathione peroxidase activity, superoxide dismutase (SOD) activity, and catalase activity. Furthermore, PSA level was utilized as an estimation of successful treatment. A marked decrease in SOD activity and a significant rise in TAC level were observed following radiotherapy in comparison with the baseline (pretreatment) values, which indicatesthe superior antioxidant activity of CUR, while no considerable alterations were seen in glutathione peroxidase activity and catalase activity. PSA levels were markedly decreased compared to baseline levels in both groups, which suggests successful treatment; however, there was no substantial difference between the two groups, which further suggeststhat CUR did not influence the efficacy of the radiotherapy [189].

In a clinical trial, Ide et al. [188] analyzed the activity of a combination of soy isoflavones and CUR on the PSA expression with increased PSA levels (but neither prostate cancer nor prostatic intraepithelial neoplasia). Altogether, 85 participants were incorporated into this study and were randomly assigned to receive either a placebo or a supplement containing a combination of CUR and isoflavones. Systematic prostate biopsy was done on the individuals prior to and six months following treatment and the PSA levels were also measured. Treatment with CUR/isoflavones markedly reduced the PSA levels in the individuals with an initial PSA ≥ 10 µg/mL in comparison with the placebo group, which they attributed to the synergistic anti-androgen activity of isoflavones and CUR [188]. Nonetheless, as there was no comparison between the effects of treatment with isoflavones alone and CUR alone compared to combination therapy on the PSA levels. Thus, more studies are required to confirm that combination therapy is more beneficial than monotherapy [188].

### 5.7. Leukemia

Leukemia is a blood cancer that mainly affects blood and bone marrow. It has been revealed that leukemia is accountable for 8% of all cancer cases (including all age groups) and involves huge costs in cases of diagnosis and treatment [328]. There are four types of leukemia including acute lymphoblastic leukemia (ALL), acute myeloid leukemia(AML), chronic lymphocytic leukemia (CLL), and chronic myeloid leukemia (CML). Patients with leukemia exhibit various common signs and symptoms including susceptibility toward various infections, weakness, fatigue, leukopenia, and anemia [329]. Furthermore, leukemia is a very common childhood cancer that accounts for around 30% of all cancers among children under the age of 15 [330]. Although marked enhancements have been achieved in leukemia therapy, the effectiveness of the available chemotherapeutic agents is still low. Leukemia involves a poor prognosis with recurrent relapses and increased mortality. It has been reported by numerous studies that the use of CUR can be beneficial in leukemia treatment.

#### 5.7.1. Anticancer Activities of Curcumin against Various Types of Leukemia

##### Acute Lymphoblastic Leukemia

Poly (ADP-ribose) polymerase-1 (PARP1) (a nuclear protein) has a significant contribution in repairing DNA damage. In addition, PARP1 plays a significant role in many pathological mechanisms including cell death, angiogenesis, cell survival, and inflammation. PARP1 overexpression has been observed in multiple primary human cancer cell lines [331,332]. In a study, Mishra et al. [333] revealed that CUR suppresses the proliferation of RS4;11 and REH cells through cleavage of PARP1 signaling pathways. It was also reported that CUR modified methylation of DNA AML cells via downregulating DNA methyltransferase 1 (DNMT1) expression, which further resulted in p15 upregulation and apoptosis of ALL cells [334]. In AML patients, the Philadelphia chromosome that encodes a fusion of breakpoint cluster region-Abelson (BCR-ABL)is a commonly observed genetic abnormality, predominantly among adults. It has been observed that this fusion contains a poor prognosis of ALL and interacts with several signaling mechanisms (such as RAF/MEK/ERK, STAT5, Akt/mTOR (mammalian target of rapamycin)), which eventually influences apoptosis of lymphoid cells. Imatinib (an inhibitor of tyrosine kinase) is typically prescribed for these patients. In these patients, Ph-positive ALL cells show poor response toward imatinib. Moreover, resistance toward imatinib is seen in these individuals [335,336]. Treatment with imatinib increases the effect of the Akt/mTOR signaling pathway, which seems to affect the efficacy of this agent. Thus, suppression of the Akt/mTOR signaling pathway might ameliorate the response toward imatinib. Indeed, CUR exhibits antitumor activities and suppresses multiple signaling cascades, particularly mTOR. Both as a single therapy and in combination with imatinib, inhibitory effects of CUR were studied on ALL cells both in vitro and in vivo. CUR also decreased the activation of ABL/STAT5 and Akt/mTOR. Furthermore, CUR decreased the *Bax/Bcl-2* ratio and downregulated BCR/ABL expression. CUR also exerted synergistic antitumor actions with imatinib in the SUP-B15 cell line, however, it suppressed cell growth in samples obtained from imatinib-resistant and recently diagnosed patients [337].

##### Acute Myeloid Leukemia

Chemoresistance is the major challenge in AML treatment. Moreover, CD34^+^ AML cells exhibit poor prognosis and resistance toward impulsive apoptosis. A study by Rao et al. (2011) studied CUR’s cytotoxic activities in DNR-insensitive CD34^+^ AML cell lines. CUR also synergistically increased the cytotoxic effects of daunorubicin and suppressed cell proliferation via arresting cell cycle in the G1/S stage. Interestingly, CUR reduced the expression of Bcl-2 mRNA and protein and stimulated caspase-3 activation. DNA methylation is mediated by DNMT1 and this enzyme catalyzes the transfer of the methyl group to DNA. Abnormal methylation of DNA can result in the silencing of tumor suppressor genes in numerous tumors and is associated with cancer pathogenesis. Thus, it is essential to develop novel inhibitors of DNA methylation with less toxicity. In a different study, Yu et al. [338] reported the in vitro and in vivo activities of CUR on the action of DNMT in AML cells. Furthermore, CUR decreased the effect of DNMT via downregulating p65 (2 positive DNMT1 regulators), NF-kB components, and Transcription Factor1 (Sp1), which further resulted in p15INK4B reactivation. Treatment with CUR markedly decreased AML tumor growth in mouse models [339].

##### Chronic Lymphocytic Leukemia

In the Western world, CLL is a very common (22–30%) form of hematological malignancy worldwide [340,341]. It has been reported that CLL-B cells interact with their microenvironment. Moreover, the survival of B cells was ameliorated via interaction with the bone marrow stromal cells. Increased lifespan of B cells caused an aberrant buildup of these cells [342]. In a study by Ghosh et al. [343], inhibitory effects of CUR on cells were obtained from CLL patients. They observed that CUR suppressed AKT, STAT3, and NF-κB signaling cascades. Treatment with the combination of CUR and epigallocatechin-3 gallate (EGCG) reversed the stromal mediated protection. In addition, CUR stimulated apoptosis through the cleavage of PARP, which further exerted an independent action on the caspase cascade [343].

##### Chronic Myeloid Leukemia

Characteristics of CML include the BCR-ABL fusion gene, which has been found to play a role in CML pathogenesis [344]. BCR-ABL contains three breakpoint cluster areas including micro (µ-BCR), minor, and major (M-BCR). It has been reported that M-BCR is the main breakpoint. Furthermore, it is responsible for encoding a 210 kDa protein and generating a 190 kDa protein, while μ-BCR encodes for a 230 kDa protein [345]. It was revealed that the P210 BCR-ABL protein plays a significant role in CML pathogenesis. In the hematopoietic system, this protein stimulated the proliferation of progenitor cells by initiating multiple cascades (such as the Ras/Raf/ MAPK pathway) and provided protection to CML cells against apoptosis [346,347]. Thus, targeting this protein is a promising therapeutic approach. Wu et al. [347] revealed the activities of CUR in K562 cells and estimated that CUR-mediated suppression of proliferation of K562 cells through p210 BCR-ABL downregulation, which further resulted in suppression of the Ras signal transduction cascade. In CML cells, Mukherjee et al. [348] reported that CUR improved the efficacy of imatinib mesylate (IM). Various doses of IM (alone) or in combination with CUR (30 μM) were used to treat K562 cells. It was indicated by the MTT assay that CUR markedly elevated the toxicity of IM. It was also exhibited by western blot analysis that when IM (alone) or acombination of IM and CUR was used, itdownregulated the expression of p210 BCR-ABL, heat shock protein 90 (Hsp90), survivin, and NF-κB subunits p65 and p50. Moreover, they reported that treatment with a combination of IM and CUR elevated the effects of *caspase-3*, *caspase-8*, and *caspase*-9 [348]. In a different study, Zhang et al. [349] showed that a combination of CUR and phosphorothioate antisense oligonucleotides led to synergistic suppressive activities on K562 cell proliferation via downregulating Hsp90, NF-κB, and P210 BCR-ABL.

## 6. Conclusions

Over the last few decades, CUR has been widely studied for its antioxidant, antiandrogenic, anti-inflammatory, and anticancer effects. Indeed, CUR has exhibited significant anticancer activities (both in vitro and in vivo) against various types of cancers including gastrointestinal, head and neck, brain, pancreatic, breast, colorectal, and prostate cancers. In various clinical studies involving human subjects, the safety and efficacy of CUR have been proven in cancer individuals either alone or in combination with other anticancer agents. CUR exerts its anticancer effect through various mechanisms including interfering with different cellular pathways and inhibiting/inducing the generation of multiple cytokines, enzymes or growth factors including IκKβ, TNF-α, STAT3, COX-2, PKD1, NFκB, EGF, MAPK, and so on. Moreover, nanoformulations of CUR can be effective in improving delivery, aqueous solubility, and efficacy compared to conventional delivery of CUR.

## Figures and Tables

**Figure 1 biomolecules-11-00392-f001:**
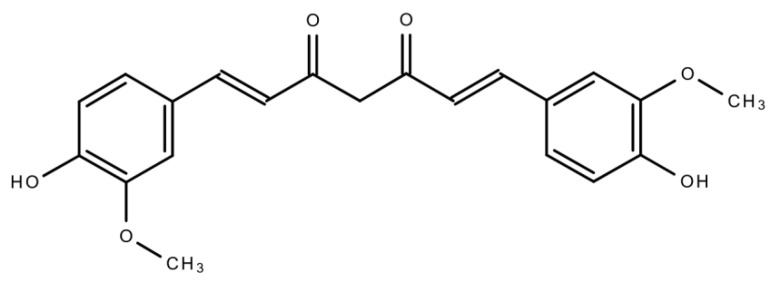
Chemical structure of curcumin.

**Figure 2 biomolecules-11-00392-f002:**
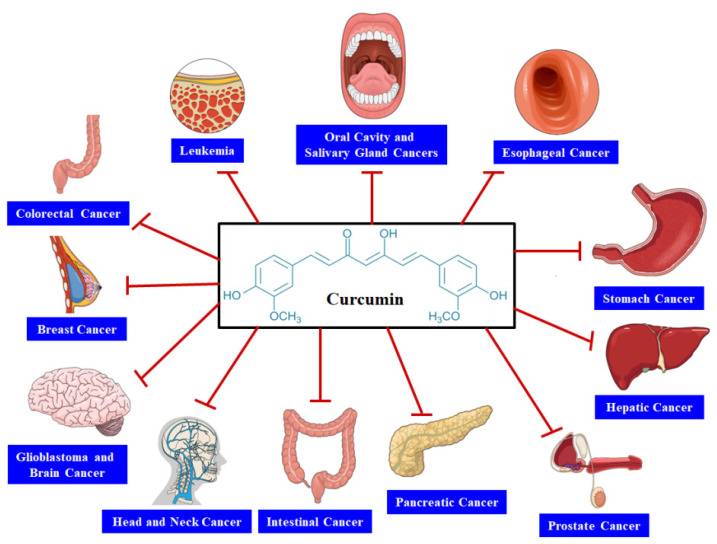
Curcumin plays a significant role in the treatment of multiple types of cancers.

**Figure 3 biomolecules-11-00392-f003:**
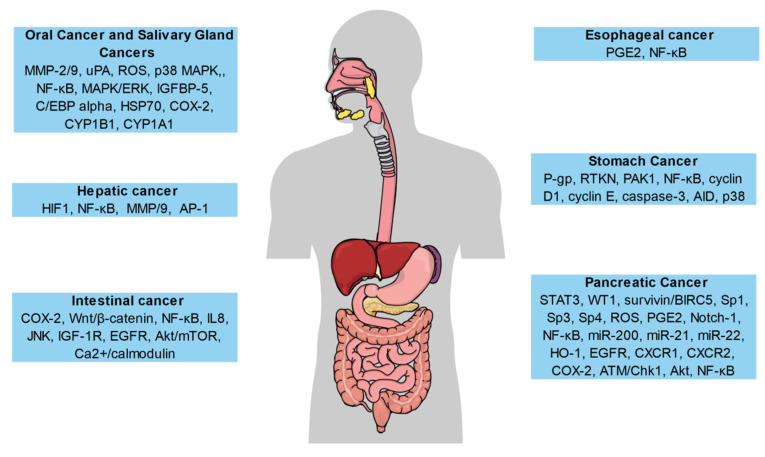
Summary of molecular targets of curcumin in gastrointestinal cancers [207]. Abbreviations: AID, activation-induced cytidinedeaminase; Akt, protein kinase B; AP-1, activated protein-1; ATM/Chk1, ataxia telangiectasia mutated/checkpoint kinase 1; BIRC5, Baculoviral IAP Repeat Containing 5; C/EBP alpha, CCAAT/enhancer-binding protein alpha; COX-2: cyclooxygenase-2; CXCR1, chemokine receptor 1; CXCR2: chemokine receptor 2; CYP1A1, cytochrome P-450 A1; CYP1B1, cytochromeP-450 B1; EGFR: epidermal growth factor receptor; HIF-1, hypoxia-inducible factor-1; HO-1, hemeoxygenase 1;HSP70, heat shock protein 70; IGF-1R: insulin-like growth factor 1 receptor; IGFBP-5, insulin-like growth factor binding protein-5; IL-1/β/8/10, interleukin 1/beta/8/10; JNK, C-jun N-terminalkinase; MAPK/ERK, mitogen-activated protein kinase/extracellular receptor kinase; miR 21/22/200, microRNA 21/22/200; MMP, matrix metalloproteinase; mTOR, mammalian target of rapamycin; NF-κB, nuclear factor kappa B; p38 MAPK, p38 mitogen-activated protein kinase; PAK1, p21-activated kinase 1; PGE2, prostaglandin E2; P-gp, P-glycoprotein; ROS, reactive oxygen species;RTKN, rhotekin; Sp 1/3/4, specificity protein 1/3/4; STAT3, signal transducers and activators of transcription 3; uPA, urokinase-type plasminogen activator; Wnt, wingless-related integration site; WT-1,Wilms’ tumor gene 1.

**Table 1 biomolecules-11-00392-t001:** Nanoformulations of curcumin and their therapeutic effects in various cancer models.

Nanoformulations	Cancer Models	Major Effects	References
Liposomes	Melanoma, colorectal cancer, and lung cancer	Enhanced bioactivity; antimelanoma effects; increased encapsulation efficiency; increased anticancer effect	[69,70,71,72,73,74,75,76]
Polymers	Colorectal cancer	Inhibited tumor growth; increased growth suppression in cancer cells as compared to free curcumin (CUR); enhanced cellular uptake; improved anticancer effect	[77,78,79,80,81]
Gold nanoparticles	Prostate and colorectal cancer cells	Ameliorated antioxidant activity; improved stability and solubility; increased biocompatibility and anticancer effect	[82,83,84]
Magnetic nanoparticles	Cancer and inflammatory cells	Ameliorated cellular uptake; potent targeting ability of CUR; controlled delivery of CUR; increased biocompatibility and anticancer activity	[85,86,87,88,89]
Solid lipid nanoparticles (SLNs)	Breast cancer lines	Prolonged blood circulation, enhanced anti-inflammatory activities; improved anticancer effect	[90,91,92,93,94,95]
Conjugates	Breast cancer	Enhanced stability, solubility, and bioavailability; potent anticancer effect	[96,97,98]
Cyclodextrins	Lung, breast, pancreatic, colorectal, and prostate cancer cells	Increased solubility, bioavailability, antiproliferation, and anticancer effects	[99,100,101,102,103,104]
Solid dispersions	Breast tumor	Prolonged survival, antitumor and anti-metastatic activity; Increased stability, bioavailability and anti-inflammatory effects	[105,106,107,108]
Micelles	Lung tumor and colorectal cancer	Improved solubility and bioavailability; extended life, targeted delivery of drug; increased chemical stability; improved anticancer and antitumor activities	[109,110,111,112,113,114,115]
Nanospheres	Breast cancer and melanoma cells	Potent antimicrobial and anticancer activities; effective targeted drug delivery	[116,117,118,119,120]
Nanogels	Colorectal cancer, pancreatic cancer and skin cancer cells	Controlled and targeted release of drug; prolonged circulation; increased bioavailability; improved anticancer activity	[121,122,123,124,125]
Nanodisks	Mantle cell lymphoma	Ameliorated biological action and apoptosis to mantle cell lymphoma and anticancer effect	[126,127,128]

**Table 2 biomolecules-11-00392-t002:** Clinical studies of curcumin in the treatment or prevention of various types of cancers.

Cancer Type	Study Type	Study Duration	Number of Participants	Outcomes	References
Breast cancer	Phase I clinical trial	7 days	14	Reduced vascular endothelial growth factor levels, decreased harmful effects, no cancer progression, partial response in some individuals	[176]
Benign prostatic hypertrophy	Pilot product evaluation study	24 weeks	61	Enhanced quality of life, decreased signs and symptoms,	[177]
Colorectal cancer	dose-escalation pilot study	29 days	15	Dose-dependently decreased the prostaglandin E2 (PGE2) levels	[178]
	Phase I dose-escalation trial	4 months	15	Lower concentrationsof curcumin (CUR) and its metabolites in urine and plasma, dose-dependently decreased the PGE2 levels	[179]
	Phase I dose-escalation trial	7 days	12	Biologically active CUR levels in the colorectal tissue	[180]
	Phase I clinical trial	30 days	126	Reduced concentrationsof tumor necrosis factor-alpha in serum, elevated *p53* expression in colorectal tissue	[181]
	Phase II clinical trial	1 month	44	Decreased number of aberrant crypt foci	[182]
	Pilot study	14 days	26	Extended levels of biologically active CUR in colon tissue, safe and well-tolerated	[183]
Chronic myeloid leukemia	Randomized controlled trial	6 weeks	50	Decreased levels of nitric oxide	[184]
Intestinal Adenoma	Randomized controlled trial	12 months	44	Very few adverse effects, no noticeable clinical response	[185]
Head and neck squamous cell carcinoma	Pilot study	-	21	Decreased activity of IκB kinase β in the salivary cells	[186]
Solid tumors	Randomized controlled trial	8 weeks	80	Enhanced quality of life, decreased inflammatory mediator levels	[187]
Prostate cancer	Randomized controlled trial	6 months	85	Reduced prostate-specific antigen levels in individuals with an initial PSA ≥ 10 µg/mL	[188]
	Randomized controlled trial	3 months	40	The considerable antioxidant effect, decreased levels of PSA	[189]
Pancreatic cancer	Phase II clinical trial	8 weeks	25	No toxicities, biological effect only in 2 individuals, poor oral bioavailability	[34]
	Phase II clinical trial	4 weeks	17	Increased incidence of side effects	[190]
	Phase I/II clinical trial	14 days	21	Safe and well-tolerated	[191]
	Phase I clinical trial	9 months	16	Enhanced quality of life, highly bioavailable, safe, no marked alterations in cytokine levels or nuclear factor kappa B activity	[192]

## Data Availability

Not Applicable.

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
