# Peer review of "Potential Role of Curcumin and Its Nanoformulations to Treat Various Types of Cancers"

_biomolecules, 2021, doi:10.3390/biom11030392_

Round 1

Reviewer 1 Report

The review by Kabir et al is focused on nanoformulations of Curcumin and their activity in several cancers. The review is well-written, and very interesting for those that work with curcumin and pharmaceutical technology. However, some sections should be more focused on the review topic. For example, some sections describe in more detail the advantages of a specific type of nanoformulation than the advantages of using that type of formulation to deliver curcumin. Moreover, some changes could be implemented before publication. I suggest that experimental values are introduced in the text to support sentences that compare the activity of curcumin with encapsulated curcumin, or to support sentences such as “excellent anticancer effect”, “outstanding anti-oxygenation activity”, etc.

Minor details:

  • Minor English changes required along the text

Author Response

Reviewer 1

Comment:

The review by Kabir et al is focused on nanoformulations of Curcumin and their activity in several cancers. The review is well-written and very interesting for those that work with curcumin and pharmaceutical technology. However, some sections should be more focused on the review topic. For example, some sections describe in more detail the advantages of a specific type of nanoformulation than the advantages of using that type of formulation to deliver curcumin. Moreover, some changes could be implemented before publication. I suggest that experimental values are introduced in the text to support sentences that compare the activity of curcumin with encapsulated curcumin, or to support sentences such as “excellent anticancer effect”, “outstanding anti-oxygenation activity”, etc.

Reviewed and Corrected:

It has been revealed by in vitro studies that CUR and its derivatives can excellently stimulate apoptosis in various cell lines through downregulating or suppressing intracellular transcription factors.

Introduction of CUR into nano-formulations to increase water-solubility has outstandingly transformed its bioavailability.

Moreover, nano-formulations have enhanced the transport and improved in vitro CUR levels in the cell, whereas their extended-release formulas as well as their increased compatibility appear to be excellent for their in vivo activities [66–68].

In MCF-7 breast cancer cells and CF-10A human mammary epithelial cells [303], excellent reduction in telomerase activity was noticed due to CUR treatment in a concentration-dependent manner which was found to be associated with hTERT downregulation via CUR instead of the c-Myc mRNA pathway [303]

Minor details:

  • Minor English changes required along the text

Reviewed and Corrected:

The authors appreciate the reviewer for the careful reviews and suggestions. As per your comments, we have deeply revised our whole manuscript.

Delivering drug at a controlled rate………… →Delivering drugs at a controlled rate……………

suppress the development of tumor→suppress the development of tumor

in the cancer treatment→in cancer treatment

studies regarding anticancer action→ studies regarding the anticancer action

An imbalance in between cell death→An imbalance between cell death

more in the track changes

Reviewer 2 Report

The authors bring a review on curcumin, its anti-cancer properties, and controllable delivery. The article is partially well written, but it lacks many important details and information such as IC50s, type of cell lines, etc.

What I value is that the authors deal not only with the anticancer potential of curcumin but also discuss poor water solubility of curcumin and its low chemical stability, which many authors in their studies or reviews omit.

Line 33: "Even though" and "but" do not really work together, try to rephrase the sentence

Curcuma longa should be in italics

curcumin (check the whole article) should not be written with capital

in the abstract, many abbreviations are not explained

Line 65: "and so on..." sounds very vague

Line 68: interest in not "on"

Chapter 2 is very brief and not very informative, it is rather a list of proteins and abbreviations, and the mechanism of curcumin action as an anticancer agent is not much described. Also, this part would benefit from some scheme or figure on the mechanism.

Line 114: add "discussed in ref. Sarco/Endoplasmic Reticulum Calcium ATPase Inhibitors: Beyond Anticancer Perspective J. Med. Chem. 2020, 63, 5, 1937–1963 https://doi.org/10.1021/acs.jmedchem.9b01509

Line 172/173 - some concentrations used? or IC50 (+ time) determined?

The same for PC-3 - statement just about what was analyzed does not say much and has no informative value

The same for line 185 + what cells and what PDT conditions, etc., as well as for all other chapters, which is rather very general, and not very informative
comparison with an effect in primary cells should be also done

Chapter 4.3 - on what basis only these conjugates were named? there are many more that were omitted and should be discussed

Chapter 5 + all subchapters - again IC50s (together with the length of the treatment) + in vitro or in vivo and the model used, must be stated

Figure 2 - caption is very poor and uninformative, it must be improved, the figure itself is of poor quality/resolution

discussion and critical view/overview of the authors on this topic is fully missing, and it has to be present

the conclusion is rather a different-way written abstract, which is unacceptable, the authors should elaborate on the conclusion and summarize the subject + possible outlook of this field

Author Response

Reviewer 2

The authors bring a review on curcumin, its anti-cancer properties, and controllable delivery. The article is partially well written, but it lacks many important details and information such as IC50s, type of cell lines, etc.

What I value is that the authors deal not only with the anticancer potential of curcumin but also discuss poor water solubility of curcumin and its low chemical stability, which many authors in their studies or reviews omit.

Line 33: "Even though" and "but" do not really work together, try to rephrase the sentence

Curcuma longa should be in italics

…….component of turmeric (Curcuma longa L.)………….

curcumin (check the whole article) should not be written with capital

Reviewed and Corrected:

For example-

CUR has been limited

CUR against hydrolysis inactivation.

CUR against several cancers

CUR in animal models

In the abstract, many abbreviations are not explained

Reviewed and corrected

IκB kinase β (IκKβ), tumor necrosis factor alpha (TNF-α), signal transducer and activator of transcription 3 (STAT3), cyclooxygenase II (COX-2), protein kinase D1 (PKD1), nuclear factor-kappa B (NF-κB), epidermal growth factor, and mitogen-activated protein kinase (MAPK)

Line 65: "and so on..." sounds very vague

so on has been deleted from line 65.

Line 68: interest in not "on"

on has been replaced with regarding

………….growing research interest regarding CUR………………..

Chapter 2 is very brief and not very informative, it is rather a list of proteins and abbreviations, and the mechanism of curcumin action as an anticancer agent is not much described. Also, this part would benefit from some scheme or figure on the mechanism.

Reviewed and corrected

In CL-5 xenograft tumors, CUR can trigger apoptosis and caused downregulation of cyclin D1, c-Met, Akt, and EGFR [27]. Furthermore, CUR suppressed metastasis and lung cell invasion via upregulating the expression of HLJ1 in cancer cells [28]. Along with the activity of CUR on NF-κB and STAT3 signaling cascades, CUR also suppressed cell cycle arrest and cell proliferation, and induced apoptosis by modulating other transcription factors including PPAR-α, Hif-1, Notch-1, β-catenin, p53, Erg-1, and AP-1 [29]. It has been confirmed that CUR suppressed the phosphorylation of focal adhesion kinase (FAK) and increased the expression of multiple extracellular matrix (ECM) components, which further contribute in metastasis and invasion. In a concentration-dependent manner, CUR also increased cell adhesion via inducing various ECM components including fibronectin, laminin, collagen IX, collagen IV, collagen III, and collagen I. Collectively, these findings have indicated that CUR inhibits FAK action via suppression of its phosphorylation sites and triggers ECM components to improve cell adhesion, which can eventually prevent cell migration and detachment of cancer cells. It was reported that suppression of FAK expression resulted in elevated cell adhesion, which eventually play role in the anti-invasive activity of CUR [30]. In colorectal cancer cells, CUR decreased the expression of CD24 in a dose-dependent manner. In addition, expression of E-cadherin was elevated by CUR and played a role as a suppressor of epithelial mesenchymal transition. In colorectal cancer cells, CUR may exhibit its action against metastasis by downregulating CD24, FAK, and Sp-1, and upregulating the expression of E-cadherin [30]. In a study, Zhou et al. [30] assessed the activity of 11 CUR-associated compounds (comprising a benzyl piperidone moiety) in various cancer cell lines. Furthermore, they observed that some of these compounds decreased the level of phospho-extracellular signal-regulated kinase (Erk)1/2 and phospho-Akt [30]. It has been reported that autophagy and ER stress might have significant contribution in case of apoptosis, which is triggered via the CUR analogue B19 in hepatocellular carcinoma cells and epithelial ovarian tumor cell line and, and that suppression of autophagy may elevate CUR analogue-triggered apoptosis via stimulating severe ER stress. In ovarian cancer cell lines, this CUR analogue might also induce apoptosis, autophagy, and ER stress in vitro[31,32]. It was confirmed that autophagy may play role in programmed cell death type II and might effectively inhibit the growth of malignant glioma cells after treatment with CUR [33].

Figure 2. Curcumin plays significant role in the treatment of multiple types of cancers.

Line 114: add "discussed in ref. Sarco/Endoplasmic Reticulum Calcium ATPase Inhibitors: Beyond Anticancer Perspective J. Med. Chem. 2020, 63, 5, 1937–1963 https://doi.org/10.1021/acs.jmedchem.9b01509.

The suggested article is not relevant to our topics at all.

Line 172/173 - some concentrations used? or IC50 (+ time) determined?

Review and corrected

Thanks for your valuable comments. We didn’t find it in line 172/173 of IC50. However, IC50 data are not available in most cases. We have added IC50 data wherever available.

In terms of cytotoxicity to Mia Paca-2 cells, Kim et al. [150] reported that paclitaxel/CUR HAS-NPs were effectively internalized into Mia Paca-2 cells and showed a 71% enhancement in IC50 versus paclitaxel HAS-NPs. Collectively, these findings indicate that paclitaxel/CUR HAS-NPs can be effectively used as anti-cancer agents in combination therapy [150].    

In the case of all the cancer cell lines, the range for IC50 of CUR-loaded PLGA nanoparticles was in between 20 μM and 22.5 μM, whereas this range for free CUR was in between 32 μM and 34 μM. Moreover, this was responsible for nearly 35% decrease in the IC50 value with CUR-loaded nanoparticles.

The same for PC-3 - statement just about what was analyzed does not say much and has no informative value.

Review and corrected

More information have been added in this regard line 236-232 and it has highlighted by yellow color.

Furthermore, expression levels of matrix metalloproteinase-2 (MMP-2)-messenger RNA (mRNA) and its proteins were detected by Reverse transcription-polymerase chain reaction (RT-PCR) and Western blotting. It was revealed that expression levels of MMP-2-mRNA and its proteins were gradually decreased along with the rise of concentrations of CUR liposomes. In PC-3 cells, it was indicated that CUR liposomes mediated intake of drug-loaded liposomes to improve the cytotoxic effects of intracellular drugs. PC-3 cells were simultaneously suppressed via the downregulation of MMP-2 concentrations.   

The same for line 185 + what cells and what PDT conditions, etc., as well as for all other chapters, which is rather very general, and not very informative comparison with an effect in primary cells should be also done.

Review and corrected

More information has been added in this regard line 243-247 and it has highlighted by yellow color.

In addition, they demonstrated that aqueous-soluble F127-CUR (a novel BLED-PDT-based system) has the capacity to mediate CUR’s anticancer effect and facilitate BLED-PDT-mediated apoptosis. Aqueous-soluble F127-CUR markedly elevated the BLED-PDT activity in the cancer cell, as compared to free CUR.

Chapter 4.3 - on what basis only these conjugates were named? there are many more that were omitted and should be discussed

Review and corrected

Piperine (an alkaloid derived from black pepper) is a strong enhancer of CUR bioavailability [154]. Furthermore, this alkaloid plays role in the brush borders of the intestinal lining, which further resulted in enhanced absorption of the compound. It has also been reported that piperine plays role in cell metabolism through via suppression of cytochrome p450s and UDP glucuronosyltransferases. Moreover, piperine exerts its action on p-glycoprotein [155]. In humans or animal models, co-administration of piperine with CUR significantly elevated the serum level of CUR by 2000-folds because of the extensive absorption and bioavailability of CUR with no adverse events [156]. Tang et al. [157] reported that poly-CURs showed cytotoxicity towards cancer cells, however, a polyacetal-based poly-CUR showed increased cytotoxicity towards MCF-7 breast cancer cell lines, and OVCAR-3, SKOV-3 ovarian cancers. Furthermore, they can be rapidly taken up by the lysosomes of cancer cells, wherein polyacetal-based poly-CUR hydrolyzed and released active CUR. In SKOV-3 cells, it arrested G0/G1 phase of the cell cycle in vitro and stimulated apoptosis of cells partly via the caspase-3 dependent cascade. In SKOV-3 intraperitoneal (i.p.) xenograft tumor model, intravenous injection of polyacetal-based poly-CUR resulted in a significant antitumor effect [157].

Chapter 5 + all subchapters - again IC50s (together with the length of the treatment) + in vitro or in vivo and the model used, must be stated.

Review and corrected

The used models or cell lines are already available in the manuscript, however more information has been provided.

CUR treatment suppressed the in vitro cell growth……………

In terms of cytotoxicity to Mia Paca-2 cells, Kim et al. [150] reported that paclitaxel/CUR HAS-NPs were effectively internalized into Mia Paca-2 cells and showed a 71% enhancement in IC50 versus paclitaxel HAS-NPs. Collectively, these findings indicate that paclitaxel/CUR HAS-NPs can be effectively used as anticancer agents in combination therapy [150].    

Figure 2 - caption is very poor and uninformative, it must be improved, the figure itself is of poor quality/resolution

Review and corrected

Figure 2 is now Figure 3 and the caption has been updated.

Summary of molecular targets of curcumin in gastrointestinal cancers

discussion and critical view/overview of the authors on this topic is fully missing, and it has to be present the conclusion is rather a different-way written abstract, which is unacceptable, the authors should elaborate on the conclusion and summarize the subject + possible outlook of this field.

Review and corrected

Information regarding CUR nanoformulations has been provided.

Moreover, nanoformulations of CUR can be effective in improving delivery, aqueous solubility, and efficacy as compared to conventional delivery of CUR.

Reviewer 3 Report

The authors summarize in this review the anticancer activity of curcumin.

Most of this review is focalized on the issue regarding the bioavailability of this natural extract. The methods used to enhance the activity of curcumin are well described but refer only to solid cancer.

1) What about curcumin action in haematological cancer?

2) Curcumin anticancer action is extensively studied, the molecular mechanism concerning cancer progression and invasiveness are well descripted. The paragraph 2 must be improving

3)The molecular pathways studied for each nanoformulation and for each type of cancer investigated, must be described

4) A figure or scheme summarizing the action of curcumin action could help the reader understand the text

Author Response

Reviewer 3

Comments and Suggestions for Authors

The authors summarize in this review the anticancer activity of curcumin.

Most of this review is focalized on the issue regarding the bioavailability of this natural extract. The methods used to enhance the activity of curcumin are well described but refer only to solid cancer.

1) What about curcumin action in haematological cancer?

Review and corrected

5.8 Leukemia

Leukemia is a blood cancer that mainly affects blood and bone marrow. It has been revealed that leukemiais accountable for 8% of all cancer cases (including all age groups) thatinvolves huge costs in cases of diagnosis and treatment [331]. There are 4 types of leukemia including Acute lymphoblastic leukemia (ALL), acute myeloid leukemia(AML), chronic lymphocytic leukemia (CLL), and chronic myeloid leukemia(CML). Patients with leukemia exhibit various common signs and symptoms including susceptibility towards various infections, weakness, fatigue, leukopenia, and anemia [332]. Furthermore, leukemia is a very common childhood cancer thataccounts for around 30% of all cancers among the children under the age of 15[333]. Although markedenhancements have been achieved in leukemia therapy, however effectiveness of available chemotherapeutic agents is still low. Leukemia involves poor prognosis with recurrent relapses and increased mortality. It has been reported by numerous studies that use of CUR can be beneficial in leukemia treatment.

5.8.1 Anticancer Activities of Curcumin against Various Types of Leukemia

5.8.1.1 Acute lymphoblastic leukemia

Poly (ADP‐ribose) polymerase‐1 (PARP1) (a nuclear protein) that has significant contribution in repairing DNA damage. In addition, PARP1 plays significant role in many pathological mechanisms including cell death, angiogenesis, cell survival, and inflammation. PARP1 overexpression has been observed in multiple primary human cancer cell lines [334,335]. In a study, Mishra et al. [336] revealed that CUR suppresses the proliferation of RS4;11 and REH cells through cleavage of PARP1 signaling pathways. It was also reported that CUR modified methylation of DNA AML cells via downregulating DNA methyltransferase 1 (DNMT1) expression, which further resulted in p15 upregulation and apoptosis of ALL cells [337]. In AML patients, the Philadelphia chromosome that encodes fusion of BCR‐ABL is a commonly observed genetic abnormality, predominantly among adults. It has been observed that this fusion contains a poor prognosis of ALL and interacts with several signaling mechanisms (such as RAF/MEK/ERK, STAT5, AKT/mTOR (mammalian target of rapamycin)), which eventually influences apoptosis of lymphoid cells. Imatinib (an inhibitor of tyrosine kinase) is typically prescribed for these patients. In these patients, Ph-positive ALL cells show poor response towards imatinib. Moreover, resistance towards imatinib is seen in these individuals [338,339]. Treatment with imatinib increases the effect of the AKT/MTOR signaling pathway that seems to affect the efficacy of this agent. Thus, suppression of AKT/MTOR signaling pathway might ameliorate the response towards imatinib. Indeed, CUR exhibits antitumor activities and suppress multiple signaling cascades, particularly mTOR. Both as a single therapy and in combination with imatinib, inhibitory effects of CUR were studied on ALL cells both in vitro and in vivo. CUR also decreased the activation of ABL/STAT5 and AKT/mTOR. Furthermore, CUR decreased Bax/Bcl-2 ratio and downregulated BCR/ABL expression. CUR also exerted synergistic antitumor actions with imatinib in the SUP‐B15 cell line, however suppressed cell growth in samples obtained from imatinib‐resistant and recently diagnosed patients [340].

5.8.1.2 Acute myeloid leukemia

Chemoresistance is the major challenge in AML treatment. Moreover, CD34+ AML cells exhibit poor prognosis and resistance towards impulsive apoptosis. In a study, Rao et al. (2011) studied CUR’s cytotoxic activities in DNR‐insensitive CD34+ AML cell lines. CUR also synergistically increased cytotoxic effects of daunorubicinand suppressed cell proliferation via arresting cell cycle in the G1/S stage. Interestingly, CUR reduced the expression of Bcl‐2 mRNA and protein and stimulated caspase‐3 activation. DNA methylation is mediated by DNMT1 and this enzyme catalyzes transfer of methyl group to DNA. Abnormal methylation of DNA can result in silencing of tumor suppressor genes in numerous tumors and is associated with cancer pathogenesis. Thus, it is essential to develop novel inhibitors of DNA methylation with less toxicity. In a different study, Yu et al. [341] reported the in vitro and in vivo activities of CUR on the action of DNMT in AML cells. Furthermore, CUR decreased the effect of DNMT via downregulating p65 (2 positive DNMT1 regulators), NF‐kB components, and Transcription Factor1 (Sp1), which further resulted in p15INK4B reactivation. Treatment with CUR markedly decreased AML tumor growth in mouse models [342].

5.8.1.3 Chronic lymphocytic leukemia

In the Western world, CLL is a very common (22–30%) form of hematological malignancy world [343,344]. It has been reported that CLL‐B cells interacted with their microenvironment. Moreover, survival of B cells was ameliorated via interaction with the bone marrow stromal cells. Increased lifespan of B cells caused aberrant buildup of these cells [345]. In a study, Ghosh et al. [346] inhibitory effects of CUR on cells obtained from CLL patients. They observed that CUR suppressed AKT, STAT3, and NF‐kB signaling cascades. Treatment with the combination of CUR and epigallocatechin‐3 gallate (EGCG) reversed the stromal mediated protection. In addition, CUR stimulated apoptosis through the cleavage of PARP, which further exerted an independent action on the caspase cascade [346].

5.8.1.4 Chronic myeloid leukemia

Characteristics of CML include breakpoint cluster region‐Abelson (BCR‐ABL) fusion gene, which has found to play role in CML pathogenesis [347]. BCR‐ABL contains 3 breakpoint cluster areas including micro (µ‐bcr), minor, and major (M‐bcr). It has been reported that M‐bcr is the main breakpoint. Furthermore, it is responsible for encodeing a 210 kDa protein and generating a 190 kDa protein, while μ‐bcr encodes for a 230 kDa protein [348]. It was revealed that the P210 BCR‐ABL protein plays significant role in CML pathogenesis. In the hematopoietic system, this protein stimulated proliferation of progenitor cells by initiating multiple cascades (such as the Ras/Raf/ MAPK pathway) and provided protection to CML cells against apoptosis [349,350]. Thus, targeting this protein is a promising therapeutic approach. In a study, Wu et al. [350] revealed the activities of CUR in K562 cells and estimated that CUR-mediated suppression of proliferation of K562 cells through p210 BCR‐ABL downregulation, which further resulted in suppression of the Ras signal transduction cascade. In CML cells, Mukherjee et al. [351] reported that CUR improved the efficacy of imatinib mesylate (IM). Various doses of IM (alone) or in combination with CUR (30 μM) were used to treat K562 cells. It was indicated by MTT assay that CUR markedly elevated the toxicity of IM. It was also exhibited by Western blot analysis that when IM (alone) or combination of IM and CUR was used, it downregulated the expression of p210 BCR‐ABL, heat shock protein 90 (Hsp90), survivin, and NF‐κB subunits p65 and p50. Moreover, they reported that treatment with combination of IM and CUR elevated the effects of caspase-3, caspase-8andcaspase-9 [351]. In a different study, Zhang et al. [352] showed that a combination of CUR and phosphorothioate antisense oligonucleotides led to synergistic suppressive activities on K562 cell proliferation via downregulating Hsp90, NF‐κB, and P210 BCR‐ABL.

2) Curcumin anticancer action is extensively studied, the molecular mechanism concerning cancer progression and invasiveness are well descripted. The paragraph 2 must be improving

Review and corrected

Furthermore, it has been demonstrated that numerous plant species exhibit anti-cancer properties and there is a growing interest regarding these plants, particularly in developing countries  [5–10].

3)The molecular pathways studied for each nanoformulation and for each type of cancer investigated must be described

Review and corrected

This information are already available in many cases. However, more information have been provided for each cancer type.

Furthermore, expression levels of matrix metalloproteinase-2 (MMP-2)-messenger RNA (mRNA) and its proteins were detected by reverse transcription-polymerase chain reaction (RT-PCR) and Western blotting. It was revealed that expression levels of MMP-2-mRNA and its proteins were gradually decreased along with the rise of concentrations of CUR liposomes. In PC-3 cells, it was indicated that CUR liposomes mediated intake of drug-loaded liposomes to improve the cytotoxic effects of intracellular drugs. PC-3 cells were simultaneously suppressed via the downregulation of MMP-2 concentrations.

It has been revealed by in vitro study that liposomal CUR treatment resulted in apoptosis [poly(ADP-ribose) polymerase] and dose-dependent growth suppression [3-(4,5-dimethylthiazol-2-yl)-5-(3-carboxymethoxyphenyl)-2-(4-sulfophenyl)-2H-tetrazolium salt] in 2 human colorectal cancer cell lines (including Colo205 and LoVo cells) [135].

In addition, they demonstrated that aqueous-soluble F127-CUR (a novel BLED-PDT-based system) has the capacity to mediate CUR’s anticancer effect and facilitate BLED-PDT-mediated apoptosis. Aqueous-soluble F127-CUR markedly elevated the BLED-PDT activity in the cancer cell, as compared to free CUR.

Khan et al. [141] demonstrated that CUR loaded-PLGA nanoparticles can significantly inhibit increased concentrations of nuclear p65 and HIF-1α in lung and breast cancer cells.  

Wang et al. [145] exhibited through Western blot analysis that Cur-SLNs mediated Bax/Bcl-2 ratio, however, reduced the expression of CDK4 and cyclin D1. These findings indicated that Cur-SLNs might be used as an effective and beneficial chemotherapeutic agent in the treatment of breast cancer [145].         

In the case of pancreatic cancer, Yallapu et al. [147] assessed the in vivo and in vitro therapeutic effectiveness of MNP-CUR formulation. Furthermore, effective internalization of MNP-CUR was observed in human pancreatic cancer cells (Panc-1 and HPAF-II) in a dose-dependent manner, which eventually resulted in effective suppression of growth of Panc-1 and HPAF-II cells in colony formation and cell proliferation assays [147].

In terms of cytotoxicity to Mia Paca-2 cells, Kim et al. [150] reported that paclitaxel/CUR HAS-NPs were effectively internalized into Mia Paca-2 cells and showed a 71% enhancement in IC50 versus paclitaxel HAS-NPs. Collectively, these findings indicate that paclitaxel/CUR HAS-NPs can be effectively used as anticancer agents in combination therapy [150].    

In a study, Kondath et al. [152] demonstrated the synergistic activity generated via gold core and CUR against breast cancer cells. They also revealed that CUR-AuNPs get coated via proteins in a biological medium, which eventually helps in their endocytosis. Furthermore, within the cells, cAuNPs induced ROS generation, which then subsequently depleted mitochondrial membrane potential. As a result, Bax was released that activated DNA fragmentation and PARP cleavage [152]. Indeed, these findings suggest the potential of CUR-AuNPs as an effective chemotherapeutic agent.

Piperine (an alkaloid derived from black pepper) is a strong enhancer of CUR bioavailability [154]. Furthermore, this alkaloid plays role in the brush borders of the intestinal lining, which further resulted in enhanced absorption of the compound. It has also been reported that piperine plays role in cell metabolism through via suppression of cytochrome p450s and UDP glucuronosyltransferases. Moreover, piperine exerts its action on p-glycoprotein [155]. In humans or animal models, co-administration of piperine with CUR significantly elevated the serum level of CUR by 2000-folds because of the extensive absorption and bioavailability of CUR with no adverse events [156]. Tang et al. [157] reported that poly-CURs showed cytotoxicity towards cancer cells, however, a polyacetal-based poly-CUR showed increased cytotoxicity towards MCF-7 breast cancer cell lines, and OVCAR-3, SKOV-3 ovarian cancers. Furthermore, they can be rapidly taken up by the lysosomes of cancer cells, wherein polyacetal-based poly-CUR hydrolyzed and released active CUR. In SKOV-3 cells, it arrested G0/G1 phase of the cell cycle in vitro and stimulated apoptosis of cells partly via the caspase-3 dependent cascade. In SKOV-3 intraperitoneal (i.p.) xenograft tumor model, intravenous injection of polyacetal-based poly-CUR resulted in a significant antitumor effect [157].

In a study, Zhang et al. [161] assessed the cellular uptake and anticancer effect of CUR-CDs. They observed that CUR-CDs improved delivery of CUR and ameliorated CUR’s in vitro and in vivo therapeutic efficacy as compared to free CUR. Thus, via regulating mitogen-activated protein kinase (MAPK)/NF-κB signaling pathway, CUR-CDs downregulated CyclinE-CDK2 combination, upregulated p53/p21 signaling cascade, and elevated the expression of Bax/caspase-3 to trigger G1-phase arrest and cellar apoptosis [161]. Collectively, these findings indicate that CUR-CDs might be used to ameliorate delivery of CUR and its therapeutic potential in case of lung cancer.

In a study, Song et al. [164] observed that solid dispersion efficiently elevated intestinal penetrability and suppressed P-gp activity. In addition, these activities elevated CUR’s anti-proliferative action in MDA-MB-231 breast cancer cells. Following two hours of incubation with CUR, solid dispersion formulation, and its physical mixture led to the differential cytotoxic activity of paclitaxel via the suppression of P-gp-induced efflux of paclitaxel in P-gp overexpressing MDA-MB-231 and LLC-PK1-P-gp cells. They also summarized that as compared to CUR, a solid dispersion formulation of CUR with mannitol and D-a-tocopheryl polyethylene glycol succinate might be a promising option for enhancing the oral bioavailability and efficacy of CUR through increased solubility, dissolution rate, cell permeability, and P-gp modulation [164].   

Li et al. [167] revealed through in vitro cytotoxicity assay that CUR micelles were reasonably more efficient as compared to native CUR against multiple cancer cell lines because of the improved cellular uptake of CUR, which further resulted in the apoptosis of cancer cell lines. In addition, increased apoptosis of S-65 cancer cells via CUR micelles was observed because of the downregulation of p-AKT, Blc-2, and p-Rb and activation of caspase-9. It was also exhibited that intraperitoneal administration of CUR micelles (25 mg/kg) may markedly suppress the growth of tumor in comparison with the treatment with native curcumin, along with a reduced expression of vascular endothelial growth factor (VEGF) in tumor tissue and markedly elevated apoptosis of tumor cells [167].

It was reported that CUR-loaded PLGA nanospheres exhibited strong intracellular uptake of the CNSs in the cells. It was also showed that CNSs exerted a strong effect on the cancer cells in comparison with free CUR [169]. In the case of all the cancer cell lines, the range for IC50 of CUR-loaded PLGA nanoparticles was in between 20 μM and 22.5 μM, whereas this range for free CUR was in between 32 μM and 34 μM.  Moreover, this was responsible for nearly 35% decrease in the IC50 value with CUR-loaded nanoparticles.

In comparison with the pure CUR and carboxymethyl cellulose and casein nanogels loaded with CUR, 2-folic acid/casein/carboxymethyl cellulose and casein nanogels loaded with CUR had decreased IC50 value and exhibited superior cytotoxic effects in MEL-39 cells owing to folate-receptor facilitated endocytosis.

As compared to control CUR, Mangalathillam et al. [175] reported that CUR loaded chitin nanogels exhibited four times more steady-state transdermal flux of CUR. They also revealed through histopathology studies that porcine skin samples which were treated with the prepared materials exhibited loosening of the outermost layer of the epidermis, which further mediated penetration along with no detected signs of inflammation. Collectively, these findings indicated that CUR-loaded chitin nanogels can be particularly used in melanoma treatment through efficient transdermal penetration [175].

In this regard, further studies indicated that these complexes triggered tumor cell apoptosis, disturbed mitochondrial membrane potential, and arrested the cell cycle in the S phase via ROS-dependent cascade.

Furthermore, they evaluated the cellular uptake of these complexes and revealed that DNA complexes containing Cu2+/Ni2+-CUR showed brighter fluorescence as compared to the complexes containing Zn2+-CUR.

4) A figure or scheme summarizing the action of curcumin action could help the reader understand the text.

Review and corrected

Figure 2. Curcumin plays a significant role in the treatment of multiple types of cancers.

Round 2

Reviewer 1 Report

I do see that the Authors ignored some of my comments that could increase the overall quality of the review. Experimental values should be added in the text to support several conclusions along the text. Anyway, the revised version of the review has improved.

Minor details:

Line 76: In instead of in

Author Response

Reviewer I

I do see that the Authors ignored some of my comments that could increase the overall quality of the review. Experimental values should be added in the text to support several conclusions along the text. Anyway, the revised version of the review has improved.

Review and corrected

We would like to take this opportunity to express our sincere gratitude to the editor for the insightful comment. According to reviewer comments we have deeply revised our manuscript point by point and it has highlighted by yellow color throughout the manuscript. We have added IC50 data wherever available. We would also like to thank you for allowing us to resubmit a revised copy of the manuscript.

In MCF-7 cancer cells, the IC50 of casein-CUR was 53 μg/ml and for casein-CUR-folic acid formulation that was used, it was decreased to 9 μg/ml, and added calcium ferrite nanoparticles exhibited little difference which resulted in the IC50 as 10 μg/ml [174].

After reference 206. The IC50 of VCR were 22.27 and 0.49 μmol/L in the resistant and sensitive lines, respectively, which suggested that the SGC7901/VCR cell line exhibited 45 folds more resistance towards VCR as compared to the parent SGC7901 cell line [3].

Cao et al. [4] reported that CUR-mediated growth inhibition in HepG2 cells was dependent on concentration and time and IC50 was 22.36 mg/ml (60.7 mM) for 48 h [256].

Khafif et al. [300] reported that radiation and CUR (range for IC50 value was 15–22 μM) suppressed cell viability in all tested cell lines. Moreover, the combination of radiation and CUR led to additive effect. CUR also reduced the expression of COX‐2 and suppressed EGFR phosphorylation in SCC‐1 cells [300]

[305]. In a study, Yin et al. [306] studied the activities of miR-326 on CUR-mediated cytotoxic effects in glioma cells. They revealed that IC50 values of CUR were 30 μM and 15 μM for U251 and U87 cells transfected with miR-326, respectively, which was significantly greater than the IC50 values observed in cells transfected with miR-326 mimics (18 μM for U251 and 9 μM for U87).

[307].In a study, Hu et al. [324] reported that CUR (with micromolar level IC50 values) markedly suppressed the proliferation of multiple breast cancer cell lines including MDA MB 468, MDA MB 231, MCF7, and T47D, which is suggesting the strong antitumor activity of CUR.

Interestingly, Teiten et al. [336] reported that androgen-sensitive prostate cancer cells showed more sensitivity towards CUR treatment (IC50 = 48 μM and 44 μM for LNCaP and 22rv1, respectively) as compared to their androgen-independent (IC50 = 170 μM and 115 μM for DU145 and PC-3 respectively) counterpart.

Reviewer 2 Report

The authors have answered my questions and improved the quality of the manuscript by additional information.

But Figure 3 is of very poor quality/resolution and also the caption is very nondescriptive, it must be improved.

Author Response

Reviewer 2

Comment:

The authors have answered my questions and improved the quality of the manuscript by additional information.

But Figure 3 is of very poor quality/resolution and also the caption is very nondescriptive, it must be improved.

According to reviewer comments we have improved our figure 3 by increasing high resolution and it has inserted in our revised manuscript.

Reviewer 3 Report

The authors answered all my questions, implementing the manuscript with several paragraphs that improved the quality of this review. 

Author Response

Reviewer 3

Comment:

The authors answered all my questions, implementing the manuscript with several paragraphs that improved the quality of this review.

Reviewed and Corrected: